# NMR-based quantification of liquid products in $CO_2$ electroreduction on phosphate-derived nickel catalysts

Phil Preikschas [1], Antonio J. Martín[1], Boon Siang Yeo [2] & Javier Pérez-Ramírez [1✉]

Recently discovered phosphate-derived Ni catalysts have opened a new pathway towards multicarbon products via $CO_2$ electroreduction. However, understanding the influence of basic parameters such as electrode potential, pH, and buffer capacity is needed for optimized $C_{3+}$ product formation. To this end, rigorous catalyst evaluation and sensitive analytical tools are required to identify potential new products and minimize increasing quantification errors linked to long-chain carbon compounds. Herein, we contribute to enhance testing accuracy by presenting sensitive $^1H$ NMR spectroscopy protocols for liquid product assessment featuring optimized water suppression and reduced experiment time. When combined with an automated NMR data processing routine, samples containing up to 12 products can be quantified within 15 min with low quantification limits equivalent to Faradaic efficiencies of 0.1%. These developments disclosed performance trends in carbon product formation and the detection of four hitherto unreported compounds: acetate, ethylene glycol, hydroxyacetone, and *i*-propanol.

---

[1] Institute of Chemical and Bioengineering, Department of Chemistry and Applied Biosciences, ETH Zurich, Vladimir-Prelog-Weg 1, 8093 Zurich, Switzerland. [2] Department of Chemistry, National University of Singapore, 3 Science Drive 3, Singapore 117543, Singapore. ✉email: jpr@chem.ethz.ch

The electrocatalytic $CO_2$ reduction reaction (eCO$_2$RR) towards valuable platform chemicals and fuels is attracting growing interest due to climate and carbon management concerns[1,2]. Current research has focused almost exclusively on copper-based catalysts owing to their propensity for C-C coupling to form multicarbon products, mainly ethylene, ethanol, and n-propanol[3,4]. Recently, we have reported that catalysts derived from inorganic Ni oxygenates (INOs; e.g., Ni phosphate, Ni carbonate, and Ni borate) can reduce $CO_2$ to linear and branched hydrocarbons up to $C_6$ and $C_1$-$C_4$ oxygenates, such as alcohols, aldehydes, ketones, and carboxylic acids[5]. With a total of 29 products, the number of reported eCO$_2$RR products has greatly increased with respect to the 16 well-established products reported for Cu-based catalysts[6–9].

Although the formation of this extended scope of products is highly promising, fundamental knowledge about the control of $C_{3+}$ product formation on INO-derived catalysts is still required before a practical application can become viable. Specifically, changing basic operating conditions, such as the applied potential, bulk pH, temperature, buffer capability, etc., can substantially change the product distribution and the overall performance toward carbon products exhibited by a catalyst[4,6,10–13]. To elucidate these critical performance trends, sensitive tools are needed to assess the complex product mixtures formed. These tools must also allow the identification of hitherto unknown eCO$_2$RR products, as a further extended scope of products cannot be excluded.

Going beyond the formation of $C_2$ and $C_3$ products adds another degree of complexity and provides new challenges in terms of product quantification. Since longer chain lengths are accompanied by an increased number of transferred electrons in eCO$_2$RR (Table 1), relative errors in product quantification can be substantially larger due to the propagation of experimental errors through common equations for determining Faradaic efficiencies (FEs). For instance, a prospective quantification error of 5 ppm in determining product concentrations would result in substantially larger relative errors for the newly observed $C_{4+}$ products (2–8 times larger) compared to those formed over Cu-based catalysts (Fig. 1 and Supplementary Note 1). Several factors can minimize or maximize this effect, such as the accuracy of quantification or not considering inlet-outlet mass flow differences. Hence, highly sensitive tools for product quantification are critically needed to minimize relative errors for long-chain products.

Whereas online gas chromatography (GC) remains the technique of choice for product quantification in the gas phase, several methods have been used to assess liquid products during eCO$_2$RR[14]. Liquid chromatography (suitable for conjugated bases of carboxylic acids, such as formate) and GC with headspace sampling ($C_1$-$C_4$ volatiles, e.g., alcohols, aldehydes, ketones) are often combined to account for the various products expected from eCO$_2$RR. While this combination offers a facile way to automatize the product analysis accompanied by a reasonably low detection limit (e.g., 20 µM for methanol)[15], product retention times are non-specific, complicating the identification of new products[14]. Nuclear magnetic resonance (NMR) spectroscopy, on the contrary, allows the simultaneous quantification of known and identification of unknown compounds once a suitable product analysis protocol is in place. In addition, it provides an even lower detection limit (<5 µM for methanol on a 400 MHz NMR spectrometer for 10 min analysis time)[15], which largely contributes to its widespread implementation in the analysis of eCO$_2$RR products[6,14,16,17]. Furthermore, with the expected improvement in catalyst functionality for chain-coupling reactions in the near future, we anticipate a further increase in the number of liquid products. These products, which will likely be a mixture of isomers (for example, $C_7H_{16}$ alone has 9 isomers), will be challenging for current gas and liquid chromatographic technologies to separate and analyze. From this perspective, we believe that it is very timely to develop a robust analytical method for quantifying liquid compounds.

Current $^1$H NMR protocols are largely confined to product mixtures usually obtained over Cu-based electrocatalysts and cannot be directly adopted to a potentially broader range of products. Notwithstanding its clear advantages, NMR spectroscopy also exhibits drawbacks such as a time-consuming and error-prone data analysis. Considering the recent progress made in high-throughput experimentation[18,19], automatization of liquid product analysis in eCO$_2$RR is highly desired to enable data-driven approaches or more advanced data analyses involving large data sets, such as machine learning[20]. Only this will allow an accelerated discovery of new materials or screening of a vast number of operating conditions.

Herein, we report refined $^1$H NMR spectroscopy protocols for assessing complex product mixtures, including an optimized water suppression using an adapted water-suppression by gradient-tailored excitation (WATERGATE) method and the reduction of NMR experiment time by adding a relaxation agent. Furthermore, combining these protocols with an Automated Product Analysis Routine (APAR) developed in Python enables substantially lower analysis times (data acquisition and processing) of less than 15 min for a sample containing up to 12 products. Using these tools alongside gas chromatography and other materials characterization techniques, we investigated a phosphate-derived Ni (PD-Ni) on carbon catalyst to identify performance trends originated from varying operating conditions (potential, conductivity, bulk pH, and buffer capacity). Through this analysis, we discovered four new products (acetate, ethylene glycol, hydroxyacetone, and i-propanol) and observed patterns, such as the different pH dependencies of oxygenate and hydrocarbon formation, and the clear alignment of methane formation with a previously proposed insertion mechanism[5]. All in all, this study profiles basic catalytic properties of PD-Ni systems. We further envision that the refined protocols and developed routines in this study can serve as a basis for the development of this new class of catalysts.

## Results and discussion

A PD-Ni catalyst was chosen as the reference catalyst in this study based on its superior selectivity towards carbon products compared to other INO-derived catalysts[5]. For this purpose, Ni phosphate supported on Vulcan XC 72 was synthesized by deposition precipitation as reported in our previous study[5] and used as a precursor for the preparation of catalyst layers coated by airbrushing on gas diffusion electrodes (GDEs)[21].

**Liquid product identification and quantification via NMR spectroscopy.** $^1$H NMR spectroscopy ranks highly among the most sensitive techniques for analyzing protonated compounds[22]. Besides its high sensitivity, $^1$H NMR spectroscopy is especially advantageous for identifying new products, as it does not necessarily require a comparison with reference samples, and can provide additional information through advanced experiments like 2D heteronuclear single quantum correlation. It can thereby be seen as a complementary technique to existing quantification methods to facilitate product identification. However, employing $^1$H NMR spectroscopy as an analytical tool for eCO$_2$RR product quantification with high precision requires the consideration of several aspects. First of all, it usually requires the use of deuterated solvents to avoid interference from NMR-active protons that belong to solvent molecules. Since aqueous electrolytes, such as dissolved $KHCO_3$ or KOH, are frequently used in eCO$_2$RR, the effectiveness of the water suppression method employed is a significant factor in the quality of the liquid product analysis.

In this work, we tailor NMR elements well established in other fields and integrate them to develop a protocol enabling highly

**Table 1 Electrochemical reactions with standard reduction potentials.**

| CN[a] | Product | Reaction | $E^{0b}$ (V vs. SHE) | Electrons transferred[c] |
|---|---|---|---|---|
| 0 | Oxygen | $2H_2O \rightarrow O_2 + 4H^+ + 4e^-$ | 1.23 | 4 |
| | Hydrogen | $2H^+ + 2e^- \rightarrow H_2$ | 0 | 2 |
| 1 | Carbon monoxide | $CO_2 + H_2O + 2e^- \rightarrow CO + 2OH^-$ | −0.93 | 2 |
| | | $CO_2 + 2H^+ + 2e^- \rightarrow CO + H_2O$ | −0.10 | |
| | Formate | $CO_2 + H_2O + 2e^- \rightarrow HCOO^- + OH^-$ | 0.28 | 2 |
| | | $CO_2 + 2H^+ + 2e^- \rightarrow HCOOH$ | −0.11 | |
| | Methanol | $CO_2 + 5H_2O + 6e^- \rightarrow CH_3OH + 6OH^-$ | −0.80 | 6 |
| | | $CO_2 + 6H^+ + 6e^- \rightarrow CH_3OH + H_2O$ | 0.03 | |
| | Methane | $CO_2 + 6H_2O + 8e^- \rightarrow CH_4 + 8OH^-$ | −0.35 | 8 |
| | | $CO_2 + 8H^+ + 8e^- \rightarrow CH_4 + 2H_2O$ | 0.17 | |
| 2 | Acetate | $2CO_2 + 5H_2O + 8e^- \rightarrow CH_3COO^- + 7OH^-$ | −0.42 | 8 |
| | | $2CO_2 + 8H^+ + 8e^- \rightarrow CH_3COOH + 2H_2O$ | 0.10 | |
| | Acetaldehyde | $2CO_2 + 7H_2O + 10e^- \rightarrow CH_3CHO + 10OH^-$ | −0.76 | 10 |
| | | $2CO_2 + 10H^+ + 10e^- \rightarrow CH_3CHO + 3H_2O$ | 0.06 | |
| | Ethylen glycol | $2CO_2 + 7H_2O + 10e^- \rightarrow (CH_2OH)_2 + 10OH^-$ | −0.57 | 10 |
| | | $2CO_2 + 10H^+ + 10e^- \rightarrow (CH_2OH)_2 + 2H_2O$ | 0.02 | |
| | Ethene | $2CO_2 + 8H_2O + 12e^- \rightarrow C_2H_4 + 12OH^-$ | −0.75 | 12 |
| | | $2CO_2 + 12H^+ + 12e^- \rightarrow C_2H_4 + 4H_2O$ | 0.08 | |
| | Ethanol | $2CO_2 + 9H_2O + 12e^- \rightarrow C_2H_5OH + 12OH^-$ | −0.74 | 12 |
| | | $2CO_2 + 12H^+ + 12e^- \rightarrow C_2H_5OH + 3H_2O$ | 0.09 | |
| | Ethane | $2CO_2 + 10H_2O + 14e^- \rightarrow C_2H_6 + 14OH^-$ | −0.69 | 14 |
| | | $2CO_2 + 14H^+ + 14e^- \rightarrow C_2H_6 + 4H_2O$ | 0.14 | |
| 3 | Hydroxyacetone | $3CO_2 + 10H_2O + 14e^- \rightarrow CH_3C(O)CH_2OH + 14OH^-$ | N.D.[d] | 14 |
| | | $3CO_2 + 14H^+ + 14e^- \rightarrow CH_3C(O)CH_2OH + 4H_2O$ | N.D.[d] | |
| | Propanal | $3CO_2 + 11H_2O + 16e^- \rightarrow C_3H_5CHO + 16OH^-$ | −0.74 | 16 |
| | | $3CO_2 + 16H^+ + 16e^- \rightarrow C_2H_5CHO + 5H_2O$ | 0.09 | |
| | Acetone | $3CO_2 + 11H_2O + 16e^- \rightarrow (CH_3)_2CO + 16OH^-$ | −0.72 | 16 |
| | | $3CO_2 + 16H^+ + 16e^- \rightarrow (CH_3)_2CO + 5H_2O$ | 0.11 | |
| | Allyl alcohol | $3CO_2 + 11H_2O + 16e^- \rightarrow CH_2CHCH_2OH + 16OH^-$ | N.D.[d] | 16 |
| | | $3CO_2 + 16H^+ + 16e^- \rightarrow CH_2CHCH_2OH + 5H_2O$ | N.D.[d] | |
| | Propene | $3CO_2 + 12H_2O + 18e^- \rightarrow C_3H_6 + 18OH^-$ | −0.73 | 18 |
| | | $3CO_2 + 18H^+ + 18e^- \rightarrow C_3H_6 + 6H_2O$ | 0.10 | |
| | Propanol | $3CO_2 + 13H_2O + 18e^- \rightarrow C_3H_7OH + 18OH^-$ | −0.73 | 18 |
| | | $3CO_2 + 18H^+ + 18e^- \rightarrow C_3H_7OH + 5H_2O$ | 0.10 | |
| | Propane | $3CO_2 + 14H_2O + 20e^- \rightarrow C_3H_8 + 20OH^-$ | −0.69 | 20 |
| | | $3CO_2 + 20H^+ + 20e^- \rightarrow C_3H_8 + 6H_2O$ | 0.14 | |
| 4 | Butadiene | $4CO_2 + 14H_2O + 22e^- \rightarrow C_4H_6 + 22OH^-$ | −0.77 | 22 |
| | | $4CO_2 + 22H^+ + 22e^- \rightarrow C_4H_6 + 8H_2O$ | 0.06 | |
| | Butanal | $4CO_2 + 15H_2O + 22e^- \rightarrow C_4H_7CHO + 22OH^-$ | −0.73 | 22 |
| | | $4CO_2 + 22H^+ + 22e^- \rightarrow C_4H_7CHO + 7H_2O$ | 0.09 | |
| | Butene | $4CO_2 + 16H_2O + 24e^- \rightarrow C_4H_8 + 24OH^-$ | −0.72 | 24 |
| | | $4CO_2 + 24H^+ + 24e^- \rightarrow C_4H_8 + 8H_2O$ | 0.11 | |
| | Butanol | $4CO_2 + 17H_2O + 24e^- \rightarrow C_4H_9OH + 24OH^-$ | −0.72 | 24 |
| | | $4CO_2 + 24H^+ + 24e^- \rightarrow C_4H_9OH + 7H_2O$ | 0.11 | |
| | Butane | $4CO_2 + 18H_2O + 26e^- \rightarrow C_4H_{10} + 26OH^-$ | −0.69 | 26 |
| | | $4CO_2 + 26H^+ + 26e^- \rightarrow C_4H_{10} + 8H_2O$ | 0.13 | |
| 5 | Pentene | $5CO_2 + 20H_2O + 30e^- \rightarrow C_5H_{10} + 30OH^-$ | −0.72 | 30 |
| | | $5CO_2 + 30H^+ + 30e^- \rightarrow C_5H_{10} + 10H_2O$ | 0.11 | |
| | Pentane | $5CO_2 + 22H_2O + 32e^- \rightarrow C_5H_{12} + 32OH^-$ | −0.70 | 32 |
| | | $5CO_2 + 32H^+ + 32e^- \rightarrow C_5H_{12} + 10H_2O$ | 0.13 | |
| 6 | Hexene | $6CO_2 + 24H_2O + 36e^- \rightarrow C_6H_{12} + 36OH^-$ | −0.68 | 36 |
| | | $6CO_2 + 36H^+ + 36e^- \rightarrow C_6H_{12} + 12H_2O$ | 0.15 | |

[a]CN: carbon number.
[b]Details on the calculation of standard reduction potentials and corresponding thermodynamic data are provided in Supplementary Note 3 and Supplementary Table 4.
[c]Number of electrons needed to form one mol of a specific compound from $CO_2$.
[d]N.D.: not determined due to missing thermodynamic data.

accurate and fast quantification of complex liquid mixtures. An adapted WATERGATE suppression method comprising a perfect echo sequence was employed to improve the water signal suppression compared to the commonly used presaturation[23–25]. WATERGATE was earlier applied in the electrocatalytic reduction of $N_2$ to ammonia and achieved a significantly enhanced signal-to-noise ratio (SNR)[24]. Despite these advantages[26,27], the permeation of this method into the field of eCO2RR still requires the development of procedures that are accessible to catalysis practitioners who may not possess advanced skills in NMR spectroscopy. To put the efficiency of WATERGATE with perfect echo in context, it enabled the detection and quantification of trace amounts of products on a state-of-the-art 500 MHz spectrometer, e.g., methanol, with a concentration below 1 μM (Table 2). Its application to a more widely available 300 MHz spectrometer provides a sufficient quantification threshold for

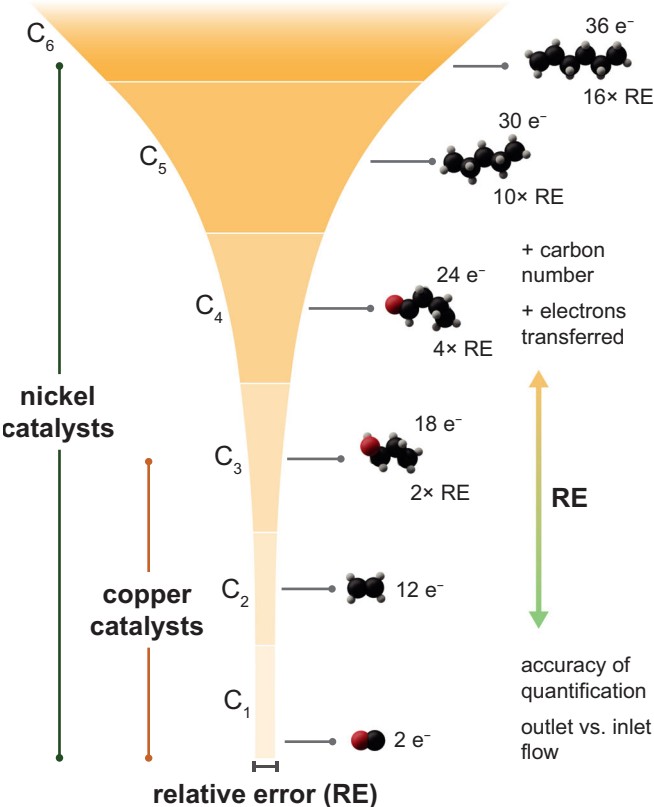

**Fig. 1 Schematic representation of error propagation in electrocatalytic CO₂ reduction reaction product analysis.** The relative error (RE) in the determination of Faradaic efficiencies FEs is directly correlated with the nature of the product, namely carbon chain length. Since the formation of long-chain products is accompanied by an increased number of electrons transferred, relative errors can be substantially increased even under small experimental errors. As a result of this error propagation, the RE for hexene exceeds 16-times the error expected for products formed by a 2 e⁻ transfer, such as CO. Not only the accuracy in product quantification but also other factors can maximize this effect, when a multi-electron transfer is involved in product formation. An Anderson–Schulz–Flory distribution with an α-value of 0.4 was assumed for error calculations. A detailed description of the error calculation is provided in Supplementary Note 1.

main products (>5 µM for methanol, Table 2) with an acquisition time of 75 min (64 scans and 60 s relaxation delay). The increased sensitivity is mainly related to improved water suppression and, ultimately, a higher receiver gain, as demonstrated for a reference sample (3 mM dimethyl sulfoxide, 60 µM formate, and 20 µM methanol) compared to spectra taken with a standard presaturation (Supplementary Fig. 1). It is worth noting that the SNRs for formate and methanol were considerably enhanced from 6.8 and 7.7 (presaturation) to 12.3 and 16.4 (WATERGATE), respectively. This is particularly significant since a reliable quantification typically requires a SNR of at least 8[24], which would not be achievable using presaturation alone in this case.

While effective water suppression enables the quantification of eCO₂RR products in general, suitable acquisition parameters must be selected. Following common guidelines of quantitative NMR (qNMR) spectroscopy[22,28,29], the relaxation delay between two excitations was identified as most significantly affecting the quantification results. In qNMR, it is widely accepted that a relaxation delay of at least five times $T_1$ (longitudinal relaxation time) of the slowest relaxing nuclei is required for a complete

signal recovery[22,28,29]. Therefore, the influence of different relaxation delays (5–60 s) was investigated on a reference sample containing nine common eCO₂RR products and dimethyl sulfoxide (DMSO) as an internal standard (for ¹H NMR spectrum, see Supplementary Fig. 2). Although shorter relaxation delays were used in other eCO₂RR studies[6,15], our investigations suggest a relaxation delay of at least 30 s to reach ≥97% signal recovery for all products investigated (Supplementary Fig. 3). For a full signal recovery, a relaxation delay of 60 s was required. These findings are consistent with previous studies reporting relaxation-derived errors in qNMR[16,30]. Bringing these errors into the context of eCO₂RR, a too short relaxation delay (e.g., 5 s) can cause an underestimation of product concentrations and, thereby, differences in FEs of up to 20–30%, such as in the case of formate, ethanol, acetaldehyde, or acetone. In addition to relaxation time, other parameters can greatly impact the product quantification, such as magnetic field homogeneity (shimming), time-domain data points, acquisition time, or post-processing parameters (further details on these parameters can be found in the Methods section).

Based on the refined NMR protocols, a combined approach of chemical shift position and coupling constant analysis, NMR spectra prediction with electrolyte shift compensation, and valida-tion with reference chemicals were employed for liquid product quantification (see Supplementary Discussion and Supplementary Fig. 4). During the catalytic evaluations of PD-Ni catalysts, we identified 14 liquid eCO₂RR products with overlapping signals in the ¹H NMR spectra, as illustrated in Fig. 2. A selection of suitable signals for reliable quantification was made and compiled in Table 2 with the respective multiplicities (additional parameters, including all possible signals, are provided in Supplementary Table 1). These tables can thus serve as the basis for product identification and subsequent quantification by using an internal standard compound such as DMSO.

**Experimental challenges and limitations of NMR spectroscopy as eCO₂RR product quantification method.** There are specific limitations of ¹H NMR spectroscopy as a quantification method in eCO₂RR. Firstly, the product of interest must contain hydrogen atoms to be probed by electromagnetic radiation, which implies that possible products without hydrogen atoms cannot be detected and quantified, such as oxalate. Another limitation is the quanti-fication of formaldehyde with a chemical shift of 4.74 ppm (as methanediol in hydrated form) overlapping with the water signal at 4.69 ppm[16]. Without a further treatment of the sample or NMR experiments at lower temperatures to exploit the temperature-dependent water chemical shift, an direct assessment of for-maldehyde is not possible[16,17]. For this reason, we performed control experiments by adding a solution of sodium bisulfite (NaHSO₃), known to form a stable formaldehyde-bisulfite adduct with an altered chemical shift[17], to the sample solutions after eCO₂RR. As no signals appeared in the respective ¹H NMR spectra at or near to the expected chemical shift of 4.58 ppm, we assumed that formaldehyde was not formed over PD-Ni or its concentration is below the detection limit.

In addition to these limitations, some experimental challenges must be considered when applying ¹H NMR spectroscopy for eCO₂RR product analysis. One important consideration is the selection of a suitable internal standard for referencing (chemical shift correction) and calibration[29]. An internal standard of a known concentration should not interfere with analytes and ideally provide a single resonance in the ¹H NMR spectrum, e.g., DMSO or dimethyl sulfone (DMSO₂). While such a reference standard accounts for differences in chemical shifts caused by the instrument or experimental conditions, pH-induced drifts must be considered

**Table 2 NMR shift positions, splitting patterns, and quantification limits of observed products.**

| CN[a] | Product | Probed nucleus | Chemical shift[b] (ppm) | Multiplicity[c] | LOQ[d] (µM) | |
|---|---|---|---|---|---|---|
| | | | | | 500 MHz[e] | 300 MHz[e] |
| 1 | Methanol | $CH_3$ | 3.23 | s | 0.7 | 6.0 |
| | Formate | $CHO^-$ | 8.33 | s | 1.9 | 16.7 |
| 2 | Ethanol | $CH_3$ | 1.06 | t | 3.0 | 29.8[f] |
| | Acetaldehyde (as diol)[g] | $CH_3$ | 1.20 | d | 3.2 | 14.3[f] |
| | Acetaldehyde | $CH_3$ | 2.12 | d | 3.2 | 14.3[f] |
| | Acetate | $CH_3$ | 1.79 | s | 0.6 | 5.7 |
| | Ethylene glycol | $CH_2$ | 3.54 | s | 0.4 | 10.0[f] |
| 3 | n-Propanol | $CH_3$ | 0.77 | t | 3.2 | 28.5 |
| | i-Propanol | $CH_3$ | 1.05 | d | 0.8 | 19.4[f] |
| | Propanal (as diol)[g] | $CH_3$ | 0.78 | t | 3.5[f] | 37.6[f] |
| | Propanal | $CH_3$ | 0.92 | t | 3.5[f] | 37.6[f] |
| | Hydroxyacetone | $CH_3$ | 2.02 | s | 1.1[f] | 12.0[f] |
| | Acetone | $CH_3$ | 2.10 | s | 0.3 | 3.0 |
| | Allyl alcohol | $CH_2$ | 3.99 | dt | 1.8[f] | 18.8[f] |
| 4 | n-Butanal | $CH_3$ | 0.97 | t | 4.4[f] | 46.6[f] |
| | n-Butanol | $CH_3$ | 0.94 | t | 4.5[f] | 47.9[f] |

[a]CN: carbon number.
[b]1H NMR chemical shifts referenced on DMSO singlet at 2.60 ppm and determined in $CO_2$-saturated 0.1 $KHCO_3$ (pH 6.8).
[c]Signal multiplicities are abbreviated as follows: s: singlet, d: doublet, t: triplet, dt: doublet of triplets.
[d]LOQ: limit of quantification based on a SNR of 6. For details on the determination of quantification limits see Supplementary Note 4 and Supplementary Fig. 6.
[e]Limits of quantification determined on a 500 MHz and a 300 MHz NMR spectrometer with the adapted WATERGATE method. Detailed information on the spectrometers used is provided in the Methods section.
[f]Limits of quantification estimated from experimental values of other products. Details on estimation are provided in Supplementary Note 4.
[g]Aldehyde and hydrolyzed form (diol) were considered for quantification.

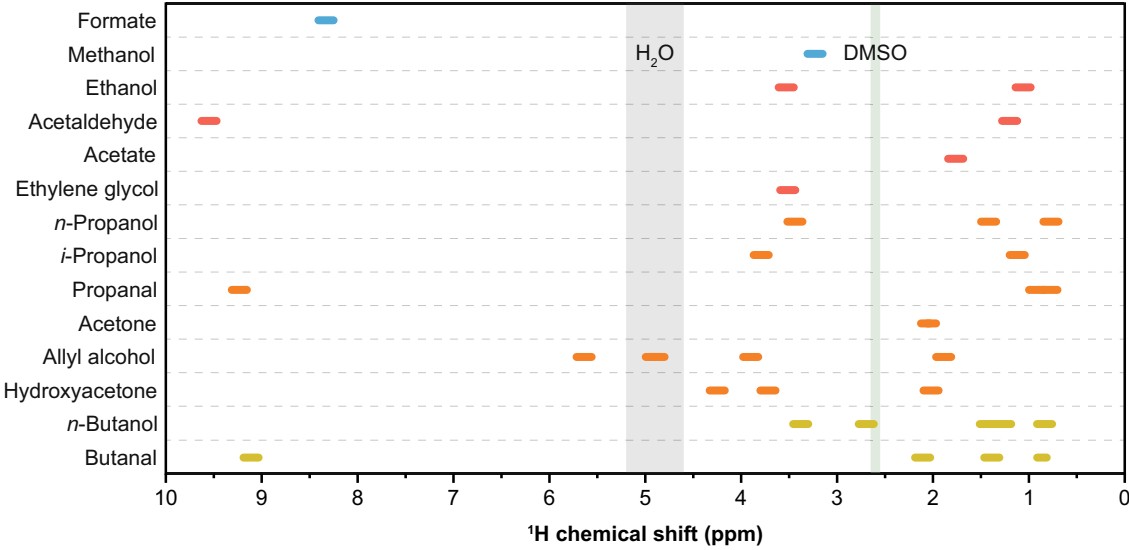

**Fig. 2 Overview of NMR chemical shift positions.** 1H NMR chemical shift positions of possible eCO2RR products over PD-Ni catalysts, water (gray), and DMSO (green; internal quantification standard). Well-separated NMR signals were selected for product quantification (Table 2).

when analyzing products obtained in different electrolytes. For example, the observed single resonance of formate ranged from 8.28 to 8.33 ppm, depending on the pH and concentration of the electrolytes used in this study (Supplementary Table 1). Another important aspect is the digital resolution, especially when quantifying product mixtures with partially overlapping signals, as indicated in Fig. 2. Among other experimental parameters (shimming, acquisition time, post-processing)[22], the digital resolution is mainly determined by the magnetic field strength of the spectrometer and the probe type used. Whereas low-field spectrometers are sufficient for quantifying products with well-separate signals, higher fields are required for complex product mixtures. To illustrate, the overlapping signals of ethylene glycol and ethanol at 3.54-3.53 ppm as

well as ethanol and i-propanol at 1.05–1.04 ppm could not be resolved on a 300 MHz spectrometer, while a sufficient resolution was obtained on a start-of-the-art 500 MHz spectrometer equipped with a cryoprobe (Supplementary Fig. 5a,b).

After considering these experimental aspects, it becomes possible to obtain low quantification limits for several eCO2RR products within reasonably short experiment times. To determine these limits, 1H NMR spectra of reference samples with concentrations ranging from 0.5 to 500 µM were recorded using the adapted WATERGATE suppression method on both a 300 MHz and a 500 MHz spectrometer. Performing a linear regression of the obtained SNRs on product concentrations resulted in limits of quantification (LOQ) for common eCO2RR products, using a SNR

threshold of 6 (regression results are provided in Supplementary Fig. 6). The LOQ values obtained ranged from 0.3 to 3.2 μM (Table 2) and were significantly lower than other reported values (5–200 μM)[15]. This improvement can be attributed to the higher sensitivity of the 500 MHz spectrometer used and the more effective water suppression by the adapted WATERGATE method. Assuming an eCO$_2$RR experiment of 60 min with a current density of 100 mA cm$^{-2}$, these product concentrations (0.3–3.2 μM) would result in FEs of 0.01–0.08%. As expected, the quantification limits obtained on the 300 MHz spectrometer were higher due to the lower sensitivity of the instrument used and ranged from 3–30 μM. Nonetheless, these product concentrations would still yield sufficient sensitivity in terms of FEs around 1%.

**Lowering NMR experiment times with MRI contrast agents.** A significant experimental challenge that needs to be addressed is the extended duration of experiments required to gather $^1$H NMR spectra for quantitative purposes[16,22,28–30]. This challenge is particularly noteworthy when considering the recent progress in high-throughput experimentation and the need for sensitive routine methods in eCO$_2$RR product quantification. These extended experiment times are mainly caused by increased relaxation delays needed to ensure full signal recovery[16]. For example, a relaxation delay of 60 s, as in this study, leads to an experiment time of 75 min (64 scans). Although correction factors can be applied to data recorded at shorter relaxation delays[22], it necessitates information on the concentration of all products of interest at partial signal recovery. A promising approach to circumvent this issue and lowering relaxation delays is the addition of an MRI (magnetic resonance imaging) contrast agent, such as Ga$^{3+}$ or Mn$^{2+}$ chelates. These paramagnetic metal salts are reported to significantly lower longitudinal relaxation times ($T_1$) and ultimately reduce relaxation delays to account for full signal recovery[16,30–33]. However, adding such an MRI contrast agent to the sample solution induces severe line broadening[16], which might hamper its application in complex product mixtures due to unresolvable, overlapping signals. Herein, ProHance®, a commercially-available Ga$^{3+}$ contrast agent, was used, and its applicability as a relaxation agent in eCO$_2$RR product quantification was investigated employing the refined $^1$H NMR protocols reported in this study.

The addition of ProHance® in a commonly reported concentration of 0.4 mM, as suggested by Hansen et al. for eCO$_2$RR product quantification[16], led to a drastic increase in peak width from 0.98 to 1.45 Hz for the single resonance of formate at 8.33 ppm (Supplementary Fig. 7a). Although the resolution obtained may be adequate for quantifying products with well-separated signals, it is necessary to identify the optimal concentration to ensure applicability for complex product mixtures produced over PD-Ni or Cu-based catalysts. Thus, the influence of different ProHance® concentrations was studied to minimize the line-broadening effect. The concentration variation yielded a pseudo-sigmoidal curve with an almost constant peak width of 0.99 Hz below 0.1 mM of ProHance® (Supplementary Fig. 7b). With this peak width, the signals of ethanol and i-propanol at 1.05–1.04 ppm could be well resolved, and the resolution is comparable with the $^1$H NMR spectrum taken without the addition of ProHance® (Supplementary Fig. 7c).

Lastly, the signal recovery of different eCO$_2$RR was investigated after the addition of 0.1 mM ProHance®. A recovery of 97% was observed for all products with a 5 s relaxation delay, and full recovery was achieved at 10 s (Supplementary Fig. 8). Therefore, the relatively long experiment times of 75 min (64 scans) could be substantially reduced by adding 0.1 mM ProHance® to 12 or 18 min, depending on the target degree of signal recovery. This approach not only decreases experiment times but also increases

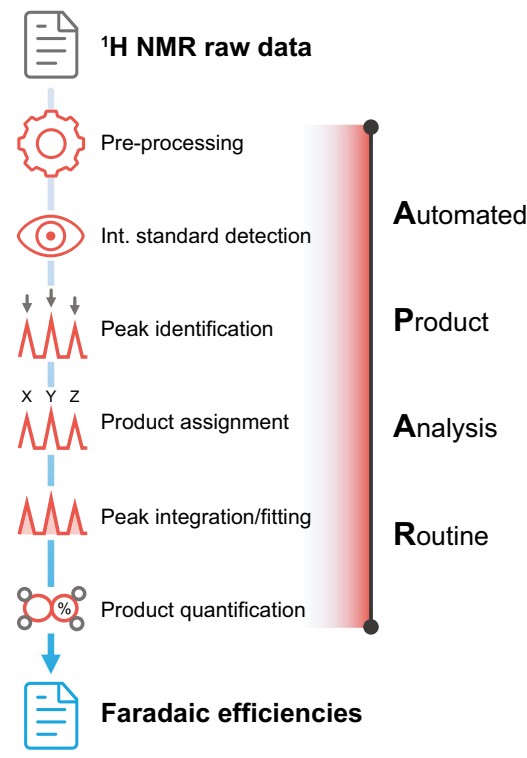

$^1$H NMR raw data

Pre-processing

Int. standard detection

**A**utomated

Peak identification

**P**roduct

X  Y  Z

Product assignment

**A**nalysis

Peak integration/fitting

**R**outine

% Product quantification

**Faradaic efficiencies**

**Fig. 3 Schematic representation of the Automated Product Analysis Routine (APAR).** Workflow of Python-based product analysis routine allowing the direct quantification of liquid products from NMR raw data of different formats. All steps are fully automatized, resulting in the calculation of product concentrations based on an internal standard. The assignment of products (step 4) is based on the chemical shift positions (Fig. 2) and coupling constants listed in Table 2. If basic electrocatalytic parameters (charged passed, CO$_2$ flow, etc.) are provided, Faradaic efficiencies are determined. Documentation and source data of the APAR are available on GitHub (https://github.com/philpreikschas/apar).

SNRs when employing similar experiment times as those without relaxation agents. Considering the expected correlation of SNR and the number of scans[22], a doubling of the SNR should be reached when increasing the number of scans from 64 to 256, which correlates with experiments times of 48 and 72 min for relaxation delays of 5 and 10 s, respectively.

**Automated liquid product analysis routine.** As relaxation agents can substantially lower experiment times, $^1$H NMR spectroscopy might be considered a prospective routine method in eCO$_2$RR product analysis. However, an automated product analysis will be required to replace the time-consuming and error-prone handling and assessment of NMR data. Although many scripts and toolboxes are available for handling NMR data and performing qNMR analysis[29], currently, none of these tools can be used for automated analysis of eCO$_2$RR products. The main reason for the limited applicability is the lack of automated identification and assignment of products.

The product-specific NMR information provided in Table 2 constituted the basis for a routine providing automated product analysis from NMR data (Fig. 3). The Automated Product Analysis Routine offered herein is written in Python and relies on open-sourced Python packages (details on the utilized packages are provided in Supplementary Note 2). The source data of APAR is available on GitHub and distributed under an open-source license. NMR raw data and an optional set of basic

electrocatalytic parameters (charged passed, $CO_2$ flow, etc.) serve as inputs for determining product concentrations and Faradaic efficiencies. Thus, the APAR can replace the time-consuming, less reproducible, and error-prone manual analysis of $^1$H NMR data for liquid product quantification. More specifically, the evaluation of a raw spectrum may take less than 1 min. In addition, the underlying table containing the product-specific NMR information can be easily edited to extend the 12 compounds already included and add information on yet-undiscovered products. A detailed description of APAR is provided in Supplementary Note 2.

This development largely extends the scope of applications in $eCO_2RR$ where NMR spectroscopy can be used as a routine product analysis method, meeting future requirements emerging from the recent progress in high throughput experimentation and the nascent integration of data science approaches in this field.

**Structural insights into PD-Ni electrodes with enhanced electrode architecture**. The pore size distribution, catalyst layer formulation, electronic and ionic conductivities, hydrophobicity, etc., will shape the properties of a GDE, requiring multidisciplinary studies towards optimized configurations[21,34–36]. However, a comprehensive electrode design will only become relevant after a catalyst formulation with practical scope has been reached. Herein, different electrode architectures were explored aiming first to maximize catalytic performance before the catalytic evaluation of different operating conditions started. Three parameters were selected: Ni content on the carbon support, the content of Nafion™ used to immobilize the catalyst particles on the gas diffusion layer, and overall catalyst content on the electrode.

A series of carbon-supported Ni phosphate precursors with different Ni contents (22.7, 37.0, and 54.0 wt%) was first immobilized on gas diffusion layers and catalytically evaluated at −1.0 V vs. reversible hydrogen electrode (RHE) in 1.0 M $KHCO_3$. This initial investigation revealed 37.0 wt% as the best performer in terms of FEs towards carbonaceous products and total current density (Supplementary Fig. 9a). This hints at possible $CO_2$ mass transport limitations faced by the excessively thick catalyst layer, which reduces catalyst overall utilization for $eCO_2RR$. Furthermore, different Ni loadings might also lead to a particle size-dependent reduction of the Ni phosphate phase. A lower reducibility could result in an increased number of sustained polarized Ni ($Ni^{\delta+}$) sites. These $Ni^{\delta+}$ sites are known to bind CO moderately, preventing the Ni surface from being poisoned once CO is formed[5]. However, further investigations in the form of *operando* X-ray absorption spectroscopy would be required to map a certain particle size effect with $Ni^{\delta+}$ site population.

Binders, such as Nafion™, impact electrode performance through various means, e.g., immobilization of catalyst particles, modulation of wettability, improvement of mass transfer and ionic conductivity[37]. Consequently, PD-Ni electrodes with different Nafion™ contents (5, 10, 18, and 28 wt%) were tested at the same conditions as described before (−1.0 V vs. RHE in 1.0 M $KHCO_3$). Among them, the best Nafion™ content was 18 wt%, leading to a substantial increase in current density by a factor of 1.5 from 26.2 (5 wt%) to 38.5 mA cm$^{-2}$ (Supplementary Fig. 9b). We assumed that this increased current density resulted from the high accessibility of protons to active sites through the Nafion™ ionomer[37–39]. A further increase in Nafion™ content slightly diminished the current density (31.8 mA cm$^{-2}$ for 28 wt%), most likely caused by the hindered mass transport of gaseous species[40,41].

Regarding the overall catalyst content on the GDEs, among a series ranging from 1.2 to 3.6 mg cm$^{-2}$, 2.0 mg cm$^{-2}$ yielded the best $eCO_2RR$ performance (Supplementary Fig. 9c). This can be

reasoned by the accessibility of active sites within the catalytic layer as a higher content, and thereby thicker layers, usually limit mass transport in GDEs[42]. For this reason, a further increase in catalyst content does not necessarily lead to a rise in available active sites and coincident reactivity. Thick catalyst layers may also involve more marked potential gradients within, decreasing energy efficiency and affecting product distribution, as it is well-known in synthesis and energy applications[43,44].

After enhancing the electrode architecture, the configuration showing the optimal performance (37 wt% Ni, 18 wt% Nafion™, 2.0 mg cm$^{-2}$ catalyst content) was structurally characterized before and after 184 min of $eCO_2RR$. Overview scanning transmission electron microscopy (STEM) of the as-prepared Ni phosphate supported on Vulcan XC 72 showed an inhomogeneous distribution of nanoparticles over the carbon support with additional formation of aggregates and agglomerates (Fig. 4a). Energy-dispersive X-ray spectroscopy (EDX) mapping of Ni and P visualizes their co-location with the nanostructured phases in the high-angle annular dark-field (HAADF) micrograph (Fig. 4b and Supplementary Fig. 10). No indications of crystalline Ni phases were obtained by STEM, even at high magnification (Supplementary Fig. 11), in agreement with the absence of corresponding reflections in X-ray diffractograms (Supplementary Fig. 12). In fact, the formation of crystalline phases is not expected, as no further heat treatment was applied after drying the catalyst precursors. It is consequently assumed that the fresh catalyst consisted of amorphous Ni phosphate phases. Therefore, FTIR spectroscopy was conducted for phase identification. The obtained spectrum showed typical bands of Ni phosphates in the fingerprint region from 1100–500 cm$^{-1}$ (Supplementary Fig. 13). However, a clear phase assignment was not possible. The reason behind this result was clarified by area-selective EDX analyses on five domains showing an average Ni:P atomic ratio of 1.3:1 (Supplementary Fig. 14), indicating that a mixture of Ni ortho- and pyrophosphate ($Ni_3(PO_4)_2$ and $Ni_2P_2O_7$, respectively) could be present, as $Ni_3(PO_4)_2$ and $Ni_2P_2O_7$ would yield ratios of 1.5 and 1.0, respectively. These findings suggest the large optimization potential lying in enhanced synthetic procedures toward better performance of PD-Ni catalysts and their fundamental understanding.

After electrocatalytic testing for $eCO_2RR$, the dispersion of Ni over the carbon support increased, as indicated by overview BF- and HAADF-STEM images showing homogenously distributed nanoparticles (Fig. 4c). Aggregation or agglomeration, as observed for the fresh sample (Fig. 4a), were not found by STEM. A first analysis of the catalyst layer structure is accessible by investigating the distribution of Nafion™. An intimate contact between Ni and Nafion™ is required to assure proton accessibility to the three-phase contact sites. STEM-EDX mapping revealed that F is in close proximity to Ni (Fig. 4d and Supplementary Fig. 15). To further elucidate the spatial distribution of Ni and F, elemental line profile analyses were performed on the Ni nanoparticles (Fig. 4d). The representative line profiles of Ni and F indicates the local enrichment of F at the Ni surface rather than an incorporation of F into the Ni structure. We, therefore, suggest that the Nafion™ binder stabilizes the Ni nanoparticles under reaction conditions and prevents agglomeration or aggregation, which might further explain the performance improvement with increasing Nafion™ content as shown in Supplementary Fig. 3b.

These structural insights highlight the complexity of active site development of INO-derived catalysts in $eCO_2RR$. Therefore, future fundamental knowledge about the crucial interplay of the active metal, support, and Nafion™ is highly desired to enable the precise tailoring of technical catalysts with improved functionalities.

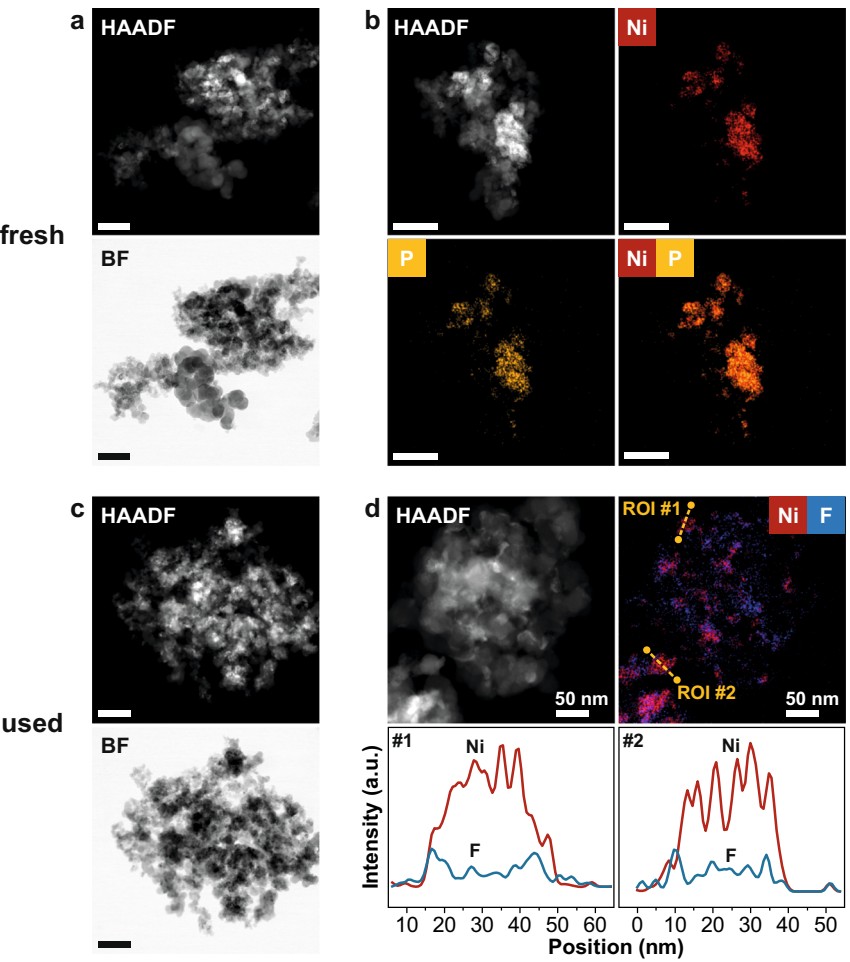

**Fig. 4 Characterization of as-prepared Ni phosphate on carbon and used phosphate-derived Ni catalyst. a** Overview HAADF- and BF-STEM images of as-prepared Ni phosphate on carbon visualizing an inhomogeneous distribution of nanoparticles and formation of aggregates/agglomerates. **b** STEM-EDX maps of as-prepared Ni phosphate on carbon indicating co-location of Ni and P from deposited Ni phosphate phase. Ni K and P K were chosen for single-element maps and superposition. **c** Overview HAADF- and BF-STEM images and (**d**) STEM-EDX maps and line-profile analysis of used PD-Ni catalyst after eCO$_2$RR for 184 min. Overview images illustrate a more homogenously distribution of nanoparticles. No indications of P was found in the respective EDX spectrum. Line profile analyses indicate the local enrichment of F at the Ni surface. Line scan paths highlighted in yellow (ROI #1 and #2) within superposition of Ni K and F K indicating good Nafion™ dispersion and contact with the active phase. Individual maps of all components and respective EDX spectra are given as Supplementary Figs. 10 and 15. Scale bars represent 200 nm unless otherwise stated.

**Performance trends in carbon product formation on PD-Ni catalyst**. Evaluating performance trends over INO-derived catalysts requires precise product analysis, as the error propagation can cause substantially increased deviations in FEs of long multicarbon products (Fig. 1). Rigorous testing protocols are indispensable toward accurate product distributions and closing of carbon balances in the eCO$_2$RR[45]. It has also been found that an unprecise determination of mass flows has a detrimental effect on the quantification of gaseous products by online GC. To illustrate, using the inlet CO$_2$ mass flow as a base for determining concentrations in the outlet stream and not considering flow alterations, such as CO$_2$ consumption or H$_2$ formation, can lead to significant overestimation of FEs of up to 12% (C$_2$H$_4$)[45,46]. Equally important, accurate potential control is a prerequisite to exploring performance trends on INO-derived catalysts since small changes in potential could significantly alter product distributions, as widely observed for Cu-based materials[4,6,47]. As highlighted in literature[11,48,49], the solution resistance might have the most significant impact on the difference between nominally applied and real potential available on the catalyst surface, requiring careful compensation of the iR drop between

the reference and working electrode. However, only 24% of a random selection of publications accounted for the iR drop and applied compensation as highlighted by a recent viewpoint[49]. Consequently, these crucial aspects were considered carefully for the following catalytic evaluations (for more details on the testing protocol, see the Methods section).

New insights into the influence of potential, bulk pH, buffer capacity, and conductivity on current density and product distribution were derived from variations of basic operating conditions: applied potential, electrolyte concentration, and electrolyte type. The derived performance trends might be of relevance in guiding the further development of INO-derived catalysts towards specific target products, e.g., increasing the oxygenate to hydrocarbon ratio.

We first explored the response of the PD-Ni catalyst to working potential by performing constant potential electrolysis. A real-time iR drop compensation was used to compensate 80% of the uncompensated resistance $R_u$ (higher compensation levels led to current oscillations), whereas a post-correction was performed to account for the remaining 20% (the suitability of this approach

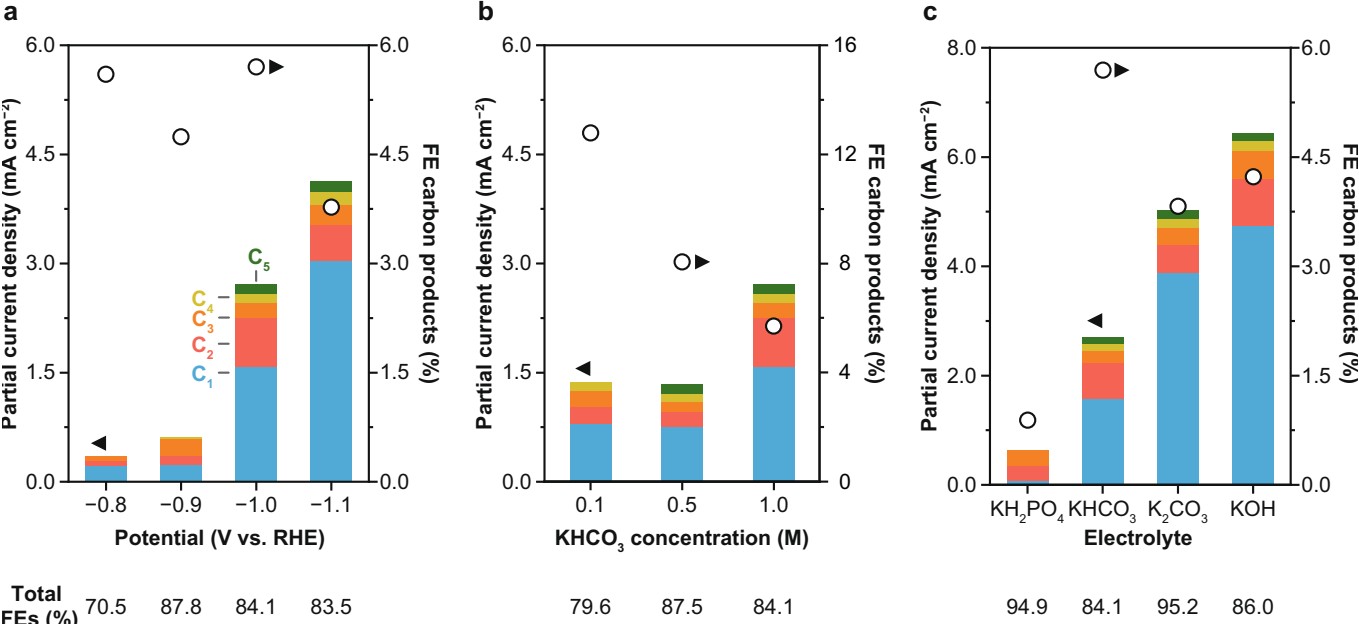

**Fig. 5 Carbon product formation on PD-Ni catalyst under different operating conditions.** Partial current densities and Faradaic efficiencies of carbon products formed over the PD-Ni catalyst grouped by carbon number under different (**a**) potentials applied, (**b**) KHCO$_3$ concentrations, and (**c**) bulk pH by using selected electrolytes. Faradaic efficiencies of all products are graphically depicted in Fig. 6 and listed in Supplementary Table 1. The remaining 65-92% of the Faradaic efficiencies must be assigned to H$_2$. Experiments were performed at −1.0 V vs. RHE and electrolyte concentrations of 1 M were used, if not specifically mentioned. Potentials were controlled with real-time *iR* drop compensation (80%). Reported values include an additional post-correction (20%). Real potentials were within 10 mV of the target values. A detailed description of the *iR* drop compensation and correction procedure is provided in the Methods section.

is shown in Supplementary Fig. 16). Figure 5a shows that increasingly cathodic potentials from −0.8 to −1.1 V vs. RHE lead to higher partial current density towards carbon products. This performance trend is mainly dominated by an increase in total current density (from *ca.* 6 to 109 mA cm$^{-2}$); a drop in the FE of carbonaceous products was observed at potentials more negative than −1.0 V vs. RHE. A Pearson's correlation analysis disclosed more specific trends in product distribution (Supplementary Fig. 17). A strong positive correlation in methane formation (Pearson Pearson's correlation coefficient $\rho$ of 0.96) with more cathodic potentials was observed, accompanied by a negative correlation in C$_{2+}$ hydrocarbon formation ($\rho = -0.82$). Both results strongly suggest that increased overpotentials favor methane formation over the C-C coupling reaction and supports the previously reported C-C coupling mechanism proposing chain growth through *CH/*CH$_2$ insertions[5]. Furthermore, it suggests that with increasing overpotentials, the fast hydrogenation of adsorbed CH$_x$ ($x$ = 1,2) species to methane results in a decreased formation of long-chain hydrocarbons. Alternatively, the suppression of the competing CO$_2$ activation reaction towards *COOH could be at play, as it is the second required surface species for C-C coupling as proposed by density functional theory (DFT)[5]. Another hypothesis is that *COOH may not only be involved in coupling reactions, but also in formate and methanol formation. Indeed, the observed trend in C$_{2+}$ product formation was accompanied by a decrease in FE towards formate and methanol, supporting this hypothesis (Fig. 6a). Analysis of the oxygenates to hydrocarbons ratio reveals that smaller overpotentials lead to increased oxygenated products (Fig. 6a and Supplementary Fig. 17).

Next, we explored the impact of local chemical environment effects inspired by the directing role they play in the formation of complex products over Cu catalysts[7]. We conducted experiments in 0.1, 0.5, and 1 M KHCO$_3$ to investigate the influence of the

underlying physicochemical parameters (buffer capacity regulating local pH and CO$_2$ concentration via equilibria) on the performance of the PD-Ni catalyst. For copper-based catalysts, the dependency of FEs on the electrolyte buffer capacity has been well-studied, demonstrating that the formation of coupled products is favored with decreased capacities (i.e., lower concentrations) linked to increased local pH values under reaction[4,50,51]. For the PD-Ni catalyst used in this study, a notable improvement in FEs towards carbon products was found with decreased KHCO$_3$ concentrations, with values ranging from 12.8 (0.1 M) to 5.7% (1 M) (Fig. 5b). However, H$_2$ remained the predominant product with FEs of 65-94% (Supplementary Table 2), formed through the parasitic hydrogen evolution reaction. Consequently, the total FEs varied between 71–95% (Fig. 5, Supplementary Tables 2 and 3), indicating an incomplete charge balance. Importantly, the FEs were calculated based on the outlet flow from the electrochemical cell, which may not align directly with studies using inlet flows for product quantification due to potentially overestimated FEs[45,46]. The incomplete carbon balance was most likely caused by liquid product and (bi)carbonate crossover, dissolved gaseous products, and quantification of volatile oxygenates. While the use of a bipolar membrane substantially lowered the product crossover compared to anion exchange membranes, a full inhibition was not achievable, in accordance with previous reports[52,53]. Ultimately, products (e.g., formate, acetate, ethanol) were found in the anolyte at concentrations of up to 5% relative to those in the catholyte. Thus, the oxidation of crossed products at the anode was inevitable. Moreover, dissolved gaseous products in the electrolyte and volatile oxygenates crossing the GDE could not be assessed, possibly contributing to the incomplete carbon balance. To this end, current GC developments and experimental setups need to be improved to minimize the potential of non-assessed products in eCO$_2$RR. Still, achieving complete charge balances in eCO$_2$RR remains a considerable

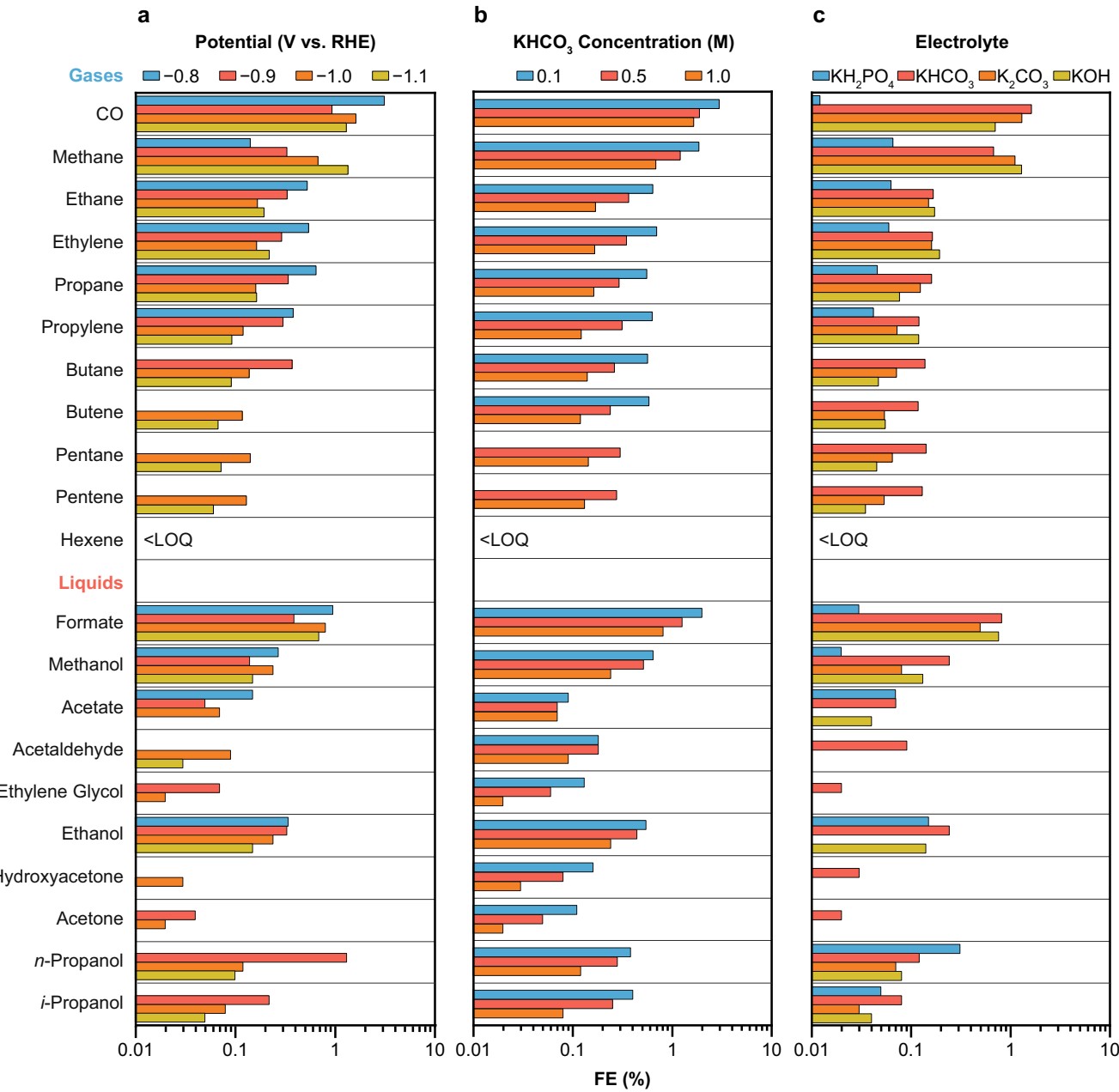

**Fig. 6 Product distribution on PD-Ni catalyst under different operating conditions.** Faradaic efficiencies of carbon products formed over the PD-Ni catalyst under different (**a**) potentials applied, (**b**) $KHCO_3$ concentrations, and (**c**) electrolytes. Experiments were performed at −1.0 V vs. RHE and electrolyte concentrations of 1 M, if not specifically mentioned. Concentration of hexene was below the limit of quantification for the GC used in this study, which is about 10 ppm. NMR signals of liquid products shown here were used for quantification when their SNR was above 6. Detailed lists of Faradaic efficiencies are provided as Supplementary Table 2 and 3. The remaining 65–92% of the Faradaic efficiencies belonged to $H_2$. Representative [1]H NMR spectra for each electrolyte used are provided as Supplementary Fig. 18.

challenge[45,46], underscoring the complexity of the process and the need for further improvements.

A similar Pearson's correlation analysis as for the variation of potentials was performed on the data obtained from different $KHCO_3$ concentrations, showing no significant difference in carbon product distributions as visualized in Fig. 6b. In fact, for all products including methane, strong positive correlations for their FEs (Fig. 7 and $\rho > 0.84$, Supplementary Fig. 17) were obtained with decreased $KHCO_3$ concentrations. This finding is in contrast to copper-based catalysts showing a product-specific change in FEs upon variation in buffer capacity[50,51]. More specifically, an increase in $C_{2+}$ product formation is accompanied by a decrease in methane

formation on Cu catalysts and vice versa. This observation underlies fundamentally different reaction mechanisms for Cu and Ni in the activation of $CO_2$ and multicarbon product formation[4,5]. While C-C bond formation on Cu is widely accepted to occur by direct coupling of adsorbed *CO species in a pH-independent step[4], INO-derived catalysts show the unique ability to form *$CH_x$ ($x = 1,2$) species responsible for C-C coupling as supported by DFT[5]. As the increase in FEs towards $C_{2+}$ products on PD-Ni was accompanied by increased methane formation, the relevance of *$CH_x$ ($x = 1,2$) species could be experimentally validated. Moreover, it underscores that the formation of methane and multicarbon products follow similar reaction pathways and

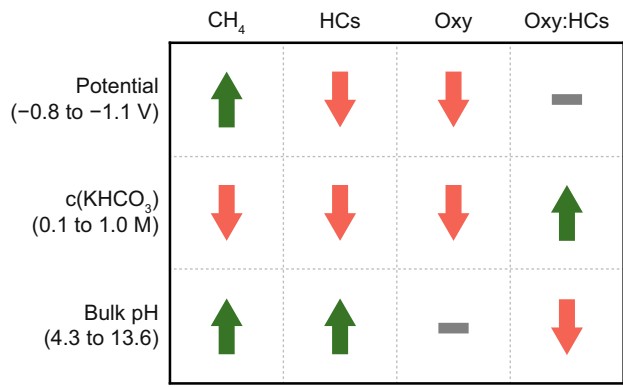

**Fig. 7 Overall trends in product formation over PD-Ni catalyst.** Arrows indicate overall trends derived from Fig. 6 and Pearson's correlation analysis (Supplementary Fig. 1).

that an apparent pH dependency exists for all products. Likewise, high concentrations of $KHCO_3$ favor oxygenate over hydrocarbon formation ($\rho = 0.99$, Supplementary Fig. 17).

Given these results, we further expanded the range of pH values by using different electrolytes from acidic (4.3, $KH_2PO_4$) over neutral (7.8, $KHCO_3$) to alkaline (11.5, $K_2CO_3$; 13.6, KOH) where carbon equilibria were largely shifted towards carbonate formation. While the highest FE towards carbon products was obtained in the 1 M $KHCO_3$ buffer, partial current densities could be substantially increased in alkaline solutions (Fig. 5c; 5.1 and 6.5 mA cm$^{-2}$ for $K_2CO_3$ and KOH, respectively). On the contrary, relatively low partial current densities were obtained in $KH_2PO_4$ (0.4 mA cm$^{-2}$), pointing at the pH-dependent nature of long-chain product formation on PD-Ni catalysts. Interestingly, a strong negative correlation appears in the alkaline pH range of 7.8 to 13.6 (Supplementary Fig. 17). Specifically, an increase in pH above 7.8 (1 M $KHCO_3$) led to decreased FEs towards long-chain products (Figs. 5c and 6c), which is masked if the full pH range is considered (Fig. 7). Assuming water as the only proton source involved in the formation of long-chain product, a direct correlation with changes in pH is to be expected. However, the change in pH alone cannot explain the different trends observed at low (4.3–7.8) and increased pH values (>7.8). For this reason, further investigations are needed to explain the observed trends and a potential role of bicarbonate ($HCO_3^-$) in the formation of carbon products on PD-Ni.

## Conclusion

The recent arrival of Ni-based catalysts enabling the formation of a variety of new multicarbon products makes the development of highly sensitive and flexible quantification techniques even more critical to achieve accurate catalyst evaluations. We established robust $^1$H NMR protocols that match these features, including optimized water suppression using an adapted WATERGATE method and a substantial reduction of NMR experiment time by adding a relaxation agent. Combining these protocols with an Automated Product Analysis Routine (APAR), which is available to all catalysis practitioners, enables the complete analysis of samples with up to 12 liquid products within 15 min and with low quantification limits (0.3–3.2 µM) correlated to Faradaic efficiencies of 0.1%. Using these tools on phosphate-derived Ni (PD-Ni) catalysts, we discovered four unreported eCO$_2$RR products (acetate, ethylene glycol, hydroxyacetone, $i$-propanol) and performance trends associated with varying potential, electrolyte buffer capability, and bulk pH. Statistical analysis revealed profound mechanistic differences in methane and long-chain product formation. While

methane formation is favored at higher overpotentials and alkaline pH values, the Faradaic efficiency towards long-chain products responds favorably to lower overpotentials and near-neutral pH. In addition, low bicarbonate concentrations promote methane and other carbon products simultaneously, whereas oxygenate formation is favored over hydrocarbons. This work lays the groundwork for facilitating the development of this new family of materials by providing sensitive and flexible tools for liquid product quantification in eCO$_2$RR, which can also be directly applied to accurately evaluate other catalysts yielding complex liquid mixtures such as copper-based ones.

## Methods

**Catalyst synthesis.** The carbon-supported nickel phosphate precursors were synthesized by a precipitation deposition method[5]. Briefly, Vulcan XC 72 was added to a solution of $NiCl_2 \cdot 6H_2O$ (50 mM) and $KH_2PO_4$ (50 mM) in ultrapure $H_2O$ (18.2 MΩ). The suspension was stirred (1000 rpm) for 30 min at r.t., before a 0.1 M KOH solution was added dropwise. After stirring for another 10 min, the mixture was filtered, and the residue was washed three times with ultrapure $H_2O$ and twice with methanol. Drying overnight in a vacuum oven (353 K, <10 mbar) provided the target product. The material was used without further treatments. The catalyst inks for gas diffusion electrodes (GDEs) were prepared by ultrasonic dispersion of the catalyst precursor (50 mg) in a mixture of ultrapure $H_2O$ (2 cm$^3$), $i$-propanol (2 cm$^3$), and 5 wt% Nafion™ solution (0.250 cm$^3$). The inks were then coated on gas diffusion layers (Sigracet 35BC, 8.8 cm$^2$ cross-sectional area) mounted on a hot plate at 353 K by airbrushing with $N_2$ as carrier gas.

**Catalyst characterization.** Scanning transmission electron microscopy (STEM) and energy-dispersive X-ray spectroscopy (EDX) were conducted on a FEI Talos F200X microscope. The microscope was operated at an acceleration voltage of 200 kV. STEM-EDX elemental maps were recorded by a SuperX system, including four silicon drift detectors. Background-corrected and fitted intensities were used for image visualization. All samples were prepared on carbon-coated copper grids. Line profile and area-selective EDX analyses were performed in Velox software (Thermo Scientific). Powder X-ray diffraction (XRD) measurements were performed in Bragg–Brentano geometry on a Rigaku SmartLab with a D/teX Ultra 250 detector using Cu $K\alpha_{1,2}$ radiation. Data were acquired in the $2\theta$ range of 10–80° with an angular step size of 0.025° and a counting time of 1.5 s per step. Fourier-transform infrared (FTIR) spectra (4000–400 cm$^{-1}$) were recorded on a Bruker Invenio-S spectrometer with a DTGS detector. Samples (approx. 10 mg) were diluted with KBr (1:10) and measured as pellets in transmission. The spectra were acquired with a spectral resolution of 4 cm$^{-1}$ and an accumulation of 32 scans with OPUS software (Bruker).

**Catalyst evaluation.** A commercially available flow cell (Micro Flow Cell, ElectroCell A/S, Denmark) with three compartments (gas, catholyte, and anolyte) was employed for all electrocatalytic experiments. The catholyte and anolyte chambers were separated by a bipolar membrane (Fumasep® FBM-PK) to minimize liquid product crossover. Complete inhibition of the product crossover was not achievable with this type of membrane. Some products (e.g., formate, acetate, ethanol) were detected in the anolyte in relatively low concentrations of up to 5% compared to the catholyte. Electrolytes were constantly circulated from separate reservoirs (40 cm$^3$) at a flow rate of 50 cm$^3$ min$^{-1}$ using peristaltic pumps. The cell was operated in a three-electrode configuration with the as-prepared GDEs as cathode (exposed area of 1 cm$^2$), a platinized Ti plate as anode, and a leak-free Ag/AgCl reference electrode (saturated, Innovative Instruments, Inc.). All experiments were conducted with an Autolab M204 potentiostat equipped with a FRA32M module for electrochemical impedance spectroscopy (EIS). CO$_2$ was constantly fed to the cell with a mass flow rate of 10 mLn min$^{-1}$ at least 30 min before the start of all experiments. The resulting outlet flow was measured with a volumetric flow meter to account for possible flow alterations and recorded for data analysis.

A typical experiment was performed in 4 sequences of EIS measurements followed by 46 min of chronoamperometry, leading to a total experiment time of 184 min. EIS (10 kHz) was employed to determine the uncompensated solution resistance $R_u$ by averaging 5 independent measurements. A real-time $iR$ drop compensation was used to compensate 80% of $R_u$ (higher compensation levels led to current oscillations), whereas a post-correction was performed to account for the remaining 20%. All potentials are reported versus the reversible hydrogen electrode (RHE) scale.

Gaseous products were analyzed every 46 min by online gas chromatography (GC) on an SRI 8610 C (Multi-Gas #3) equipped with one thermal conductivity and one flame ionization detector with methanizer. Products were separated on HayeSep D and Molecular Sieve 13X packed columns using Ar as carrier gas. This GC configuration allowed the separation of small molecules (H$_2$, CO, CO$_2$, CH$_4$), C$_1$–C$_6$ alkanes, and C$_2$–C$_6$ alkenes without isomers. Faradaic efficiencies were calculated based on the recorded outlet flow, GC product concentrations, and the

current at the sampling time. Liquid products were quantified after reaction by [1]H NMR spectroscopy.

**[1]H NMR data acquisition and processing**. 1D [1]H NMR spectra were recorded on a Bruker AVANCE III HD spectrometer equipped with a 11.75 T magnet ([1]H Larmor frequency of 500 MHz) and a 5 mm BBO Prodigy CryoProbe (Bruker BioSpin) abbreviated as "500 MHz spectrometer" or a Bruker AVANCE III HD spectrometer equipped with a 7.0 T magnet (300 MHz) and a 5 mm BBFO probe abbreviated as "300 MHz spectrometer". The frequency lock was set for 10% $D_2O$ and 90% $H_2O$ and, followed by phase and shim adjustments. The data was acquired using a perfect echo W5 WATERGATE solvent suppression pulse sequence (adapted PEW5) with the transmitter frequency offset (o1p) centered at water signal (4.7 ppm)[23–25]. For product identification, 512 scans were accumulated with a relaxation delay (d1) of 5 s, size of FID (td) of 32k datapoints, and an acquisition time (aq) of 3.3 s. In addition, a presaturation sequence (zgpr) was employed for comparative studies. Data was acquired with Bruker TopSpin and processed (apodization with lb value of 0.2 Hz, zero filling, phasing) with Mnova software suite.

**Liquid product quantification**. 1D [1]H NMR spectra for quantitative purposes were recorded on the 500 MHz spectrometer using the adapted WATERGATE suppression method. 64 scans were accumulated with a relaxation delay (d1) of 60 s, size of FID (td) of 64k datapoints, and an acquisition time (aq) of 6.4 s. For sample preparation, an aliquot (0.54 cm$^3$) of the respective catholyte was mixed with a dimethyl sulfoxide (DMSO) solution in $D_2O$ (0.06 cm$^3$, 0.5 mM). $D_2O$ was used for frequency locking, and DMSO served as internal referencing and quantification standard. Samples were measured without any further treatment. Post-reaction elemental analysis via inductively coupled plasma optical emission spectroscopy (ICP-OES) was performed to confirm the absence of paramagnetic, high-spin $Ni^{2+}$ species which could potentially impact the [1]H NMR data acquisition. Product concentrations were determined based on the six equivalent hydrogen atoms of DMSO and the integral area of its single resonance at 2.60 ppm. Data was processed (apodization with lb value of 0.2 Hz, zero filling, phasing) using the Automated Product Analysis Routine (APAR) reported in this study.

To account for the possible formation of formaldehyde, a solution of sodium bisulfite (NaHSO$_3$, 5 M, 1 cm$^3$) was well-mixed with an aliquot (1 cm$^3$) of the respective catholyte. [1]H NMR spectra were then recorded from the resulting mixtures (0.54 cm$^3$) after the addition of a dimethyl sulfone (DMSO$_2$) solution in $D_2O$ (0.06 cm$^3$, 0.5 mM).

**[1]H NMR signal recovery investigations**. 1D [1]H NMR spectra for signal recovery investigations were recorded on the 500 MHz or 300 MHz spectrometer using the adapted WATERGATE suppression method. 64 scans were accumulated with relaxation delays (d1) of 0.1–60 s, size of FID (td) of 64k datapoints, and an acquisition time (aq) of 6.4 s. For investigations regarding the addition of an MRI relaxation agent, the reference sample (Supplementary Fig. 2) was mixed with 0.036, 0.048, 0.054, or 0.057 cm$^3$ of a DMSO solution in $D_2O$ (0.5 mM) and 0.024, 0.012, 0.006, or 0.003 cm$^3$ of a ProHance® solution in $D_2O$ (10 mM) yielding ProHance®-concentrations of 0.4, 0.2, 0.1, and 0.05 mM.

## Data availability
Data presented in the main figures of the manuscript and NMR data are publicly available through the Zenodo repository (https://doi.org/10.5281/zenodo.7848651). Further data supporting the findings of this study are available in the Supplementary Information. All other relevant source data are available from the corresponding author upon reasonable request.

## Code availability
Source data of the Automated Product Analysis Routine (APAR) reported in this study is open-sourced on GitHub (https://github.com/philpreikschas/apar) and additionally available as Zenodo repository (https://doi.org/10.5281/zenodo.8070371).

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

## Acknowledgements

This publication was created as part of NCCR Catalysis (grant number 180544), a National Center of Competence in Research funded by the Swiss National Science Foundation. B.S.Y. thanks the National Research Foundation of Singapore (Urban Solutions and Sustainability, Industry Alignment Fund (Pre-Positioning) Program, A-0004543-00-00) for financial support. The Scientific Center for Optical and Electron Microscopy (ScopeM) at the ETH Zurich is thanked for access to their facility. Mr. Dario Faust Akl is thanked for acquiring the EM data, and Dr. René Verel for assistance with NMR data acquisition.

## Author contributions

J.P.-R. and B.S.Y. conceived and coordinated the study. P.P., A.J.M., and J.P.-R. conceptualized the study, P.P., A.J.M., B.S.Y., and J.P.-R. wrote the article. P.P. synthesized the catalysts, contributed to their characterization, conducted catalytic tests, and performed NMR experiments.

## Competing interests

The authors declare no competing interests.
