## [Peer Review File · Communications Chemistry]

Reviewers' comments:

Reviewer #1 (Remarks to the Author):

The work of Preikschas and co-workers reports CO₂ electroreduction on phosphate-derived Ni catalysts. Using a ¹H NMR protocol and automated data analysis the authors study the selectivity of these catalysts under different reaction conditions, and are able to detect four new products for this reaction. The work is interesting, and I recommend it to be published after major revisions are made. I am missing details on the detection limitations for the different products depending on spectrometer used. It is unclear if this is intended for being a method development or catalyst development paper. At the moment it is neither. The catalyst used makes >90% hydrogen regardless the reactions conditions. The authors say their work provides the basis for developing this catalyst further for targeting specific products, however I do not see this "basis" in the manuscript. What it is provided is a methodology for analysis products – which is great – but not really specific guidelines for improving this catalyst (I mean a real improvement, not from 2 to 4% FE). I suggest the authors rephrase parts of the manuscript and focus more on the method development – which is the strong message in this manuscript. Avoid overselling. Please see below more specific comments:

1- I suggest the authors make the title more specific. For example:

"Multicarbon product quantification via ¹H NMR to evaluate CO₂ electroreduction over phosphate-derived nickel catalysts"

2- Please avoid overselling the work as "outstanding" as in the last sentence in the abstract: "This work provides an outstanding practical platform for catalyst development." Let the work speak for itself. ¹H NMR with an automated analysis protocol is not necessarily practical for every group working on CO₂ electroreduction.

3- Check the English throughout the whole manuscript. It can be considerably improved to ease the article readability.

4- The authors mention the quantification threshold for main products and give the example of methanol (>5 μM with a 300 MHz spectrometer, below 1 μM with a 500 MHz spectrometer). Please provide the same information for other common CO₂ reduction products. This will significantly increase the impact of your work and accessibility.

5- Can the authors please discuss the challenges and limitations of the technique for CO₂RR product analysis more? What can affect chemical shifts? What resolution (ppm) you have? What determines that? What are the experimental challenges and "details"? That is largely missing in the manuscript.

6- Please include a brief description of the error propagation calculation from Fig 1 also in the main text. Parts of the caption of Fig 1 can in fact move to the main text, to make it easier to understand when the authors refer to it.

7- Why do the authors choose to only correct for 80% of the solution resistance during the experiment? Why not 85%? 90%?

8- The plural of pH is pH. Not pHs.

9- Please write down in the caption of Figures 5 and 6 what product is responsible for the other ~80-90% of faradaic efficiency (I assume H₂?)

10- In the conclusions the authors write "This work sets the practical basis for developing this new family of materials towards desired target products." However, the main product this catalyst make under CO₂ atmosphere is still hydrogen >90% for all conditions tested. This manuscript is nice as a

“technical development” in the sense of the product detection methodology presented. However, as for the development of CO₂ electrocatalysts, I am missing a discussion as to how to stir the selectivity of these Ni-based catalysts, and how to improve the C-products faradaic efficiency. Without it, the statement in the conclusions is vague – what basis is being set for developing this new family of materials towards target products? As mentioned before: avoid overselling your work.

Reviewer #2 (Remarks to the Author):

Preikschas et al provide a carbon identification and quantification approach based on NMR Spectroscopy for analyzing the substrate mixture after CO₂ electroreduction over Ni catalysts as described in reference 5. The context of advanced analytics to this latter paper which has met much interest in the reception leave no doubt that this is a timely and relevant paper. I think, however, that the work would make for a much more compelling story that raises about the feel of additional analysis of previous work, if the following could be considered:

1. The analytical part and the application part could be tied better together in my opinion, especially in the results section some transition from method to application would be desirable.

In the application part, phrasing could be more accurate, for instance it cannot be any surprise that acetate is found in this reaction, albeit it has not been identified in reference 5. This part culminates in speculation that is not discussed in depth, that bicarbonate and other buffer anions contribute beyond a pH effect, where bicarbonate serves as a hydrogen donor. Meaningful controls (if possible, at all) or computational work to corroborate the hypothesis would be desirable, as analogy to Au and Ag catalysts do not end the results section on a very rigorous note. A schematic figure of the suggested mechanistic effects including the newly identified chemical could be beneficial and improve impact. Overall and elsewhere, Figures are very nice.

I would prefer if the conclusion could end on a less generic note than “This work sets the practical basis for developing this new family of materials towards desired target products.”-why not mention which specific questions have been addressed and which ones remain?

2. This work is not least presented as an analytical toolbox, but many details are hard to judge, despite if Figure 3. The scripts are referred to as being available on github and the link should probably be repeated under (Supplementary Note 3). Does the toolbox provide a deconvolution and peak fitting in addition to integration and how are pH dependent shift changes addressed? Is the approach novel - similar approaches and toolboxes must exist in the metabolomics world? How is peak overlap addressed, for instance Figure 2 indicates that ethylene glycol will only yield overlapped signal?

3. Experimental details about the NMR method remain a bit unclear to me:

A) Is the improved limit of detection the consequence of a bigger receiver gain owing to better water suppression? Is sensitivity really better than for GCMS? Could presaturation be optimized (e.g., on a standard, non-Cryoprobe instrument that may be accessible to most users) to allow for similar receiver

gains as the WATERGATE experiment?

B) Acquisition times and recycle delays should be provided. Is it correctly understood that quantifications are achieved relative to internal DMSO by considering the six equivalent hydrogens from the internal standard? Can you show real spectra of the analyzed samples rather than solely the reference sample of SI1?

C) How is the different relaxation behavior for different mixture components and the need to wait for $5 \cdot T_1$ accounted for in the quantification? How can you be sure that water suppression does not suppress signals near the water, for instance? Does the presence of paramagnetic species in samples need to be considered? And are samples neutralized before analysis to facilitate peak-ID?

D) Is there any attempt to quantify formaldehyde or is the method agnostic to formaldehyde (see e.g., ref 17)?

E) How is the presence of different forms accounted for, for instance acetaldehyde should be present in hydrate and aldehyde form, while only the hydrate form seems to be considered. Also, propionaldehyde likely may exist in a hydrate form, while here only the aldehyde form seems to be considered.

F) Can you provide error estimates for your quantifications?

G) In Figure 2-why are there two NMR signals for methanol (I assume the OH would exchange quickly and only the methyl group is visible, also the shift seems not in accordance with hydroxy groups); probably "1H" Chemical shift should be named in the figure legend.

4. Some minor textual issues can be easily fixed:

"till" throughout change to "until"

page 4 C7H14 should be C7H16

page 5 I am not sure if it is advisable to speculate on the analysis time of 20-60 min without the method, without any further substantiation

tempus: past tense throughout results section is encouraged.

Ideally avoid using "This" without a noun; especially "This underlies fundamentally different reaction mechanisms for Cu and Ni in the activation of CO₂ and multicarbon product formation." becomes a suboptimal sentence, even if the meaning is understandable enough (This observation is consistent with previously suggested different reaction...)

Ideally avoid starting sentences with "To illustrate," and "On the contrary" (By contrast)

Extra section for NMR data acquisition under methods is advised, including recycle delay, pH, experiment times, acquisition time of FID etc; maybe consider changing volumes from cm³ to mL.

Table 1 could benefit from spaces between stoichiometric coefficients and chemicals/electrons

COMMSCHEM-23-0058 - Response to Reviewers

Comments in *blue* | Replies in black | Actions in **bold**.

Reference to figures, page and line numbers refer to the manuscript with highlighted changes.

Reviewer #1

The work of Preikschas and co-workers reports CO₂ electroreduction on phosphate-derived Ni catalysts. Using a ¹H NMR protocol and automated data analysis the authors study the selectivity of these catalysts under different reaction conditions, and are able to detect four new products for this reaction. The work is interesting, and I recommend it to be published after major revisions are made. I am missing details on the detection limitations for the different products depending on spectrometer used. It is unclear if this is intended for being a method development or catalyst development paper. At the moment it is neither. The catalyst used makes >90% hydrogen regardless the reactions conditions. The authors say their work provides the basis for developing this catalyst further for targeting specific products, however I do not see this “basis” in the manuscript. What it is provided is a methodology for analysis products – which is great – but not really specific guidelines for improving this catalyst (I mean a real improvement, not from 2 to 4% FE). I suggest the authors rephrase parts of the manuscript and focus more on the method development – which is the strong message in this manuscript. Avoid overselling. Please see below more specific comments:

We thank the Reviewer for their constructive and valuable feedback. In response to the main suggestion, we have revised our manuscript to emphasize the value added by the analytical development and to underscore its critical role in catalyst development for this new family of materials. To this end, we now provide a more comprehensive description of the NMR method, including clear indications of its scope, limitations, and practical application. We would like to also emphasize that we are embarking on the development of a novel class of catalyst, which could reduce CO₂ to long-chain hydrocarbons. Even though its Faradaic efficiencies (*FEs*) are currently low compared to the well-studied and optimized Cu system, it is our belief that our Ni-based electrocatalysts can be further improved in terms of *FEs* and overall performance. Given its unprecedented ability to form C₃₊ products from eCO₂RR, we are convinced that fostering the development of Ni-based electrocatalysts is of utmost relevance and that the developed NMR method can further facilitate research on these catalysts. We also clarify that the observed hydrogen *FEs* were between 68-94% and are influenced by the specific operating conditions. Moreover, the *FEs* towards carbon products ranged from 1.5-12%, and with that, the overall reactivity in terms of partial current density was improved by a 4-fold increase (0.1 M KHCO₃ vs. 1 M K₂CO₃).

1. I suggest the authors make the title more specific. For example: “Multicarbon product quantification via 1H NMR to evaluate CO₂ electroreduction over phosphate-derived nickel catalysts”

The title has been changed to “Quantification of multicarbon CO₂ electroreduction products formed on phosphate-derived nickel catalysts” as requested by the Reviewer. We believe this title conveys the key messages of the revised manuscript now focusing on method development.

2. Please avoid overselling the work as “outstanding” as in the last sentence in the abstract: “This work provides an outstanding practical platform for catalyst development.” Let the work

speak for itself. ¹H NMR with an automated analysis protocol is not necessarily practical for every group working on CO₂ electroreduction.

We thank the Reviewer for this critical comment. The adjective “outstanding” served as a modifier of the term “practical platform” and was not intended to describe the entire work. However, we agree that this phrase may be ambiguous and **have deleted sentences including this adjective on pages 2 and 16**. Furthermore, **we have thoroughly checked the entire manuscript to nuance language**.

Regarding the Reviewer's comment on the practicality of using ¹H NMR spectroscopy for product quantification in CO₂ electroreduction, we agree that this method might not be practical for every research group. Nevertheless, we would like to emphasize that our protocols and the automated product analysis routine were not developed to replace existing quantification methods but rather to complement them. Additionally, ¹H NMR spectroscopy is especially advantageous for identifying new products, as it does not require a comparison with reference samples (necessary when using gas or liquid chromatography) and can also provide additional information through advanced experiments like 2D HSQC (heteronuclear single quantum correlation). **We have clarified this aspect further on page 6, line 6**. We are convinced that considering the potential for an expanded scope of products, ¹H NMR spectroscopy may be particularly relevant for eCO₂RR product analysis aiding the development of novel catalytic systems targeting multicarbon compounds.

3. Check the English throughout the whole manuscript. It can be considerably improved to ease the article readability.

We have revised the original manuscript to improve its readability.

4. The authors mention the quantification threshold for main products and give the example of methanol (>5 μM with a 300 MHz spectrometer, below 1 μM with a 500 MHz spectrometer). Please provide the same information for other common CO₂ reduction products. This will significantly increase the impact of your work and accessibility.

We performed a four-fold serial dilution of a reference sample containing common CO₂ electroreduction products with concentrations ranging from 0.5 to 500 μM. 1D ¹H NMR spectra of the resulting solutions were recorded on both a 300 MHz and a 500 MHz spectrometer using the adapted WATERGATE suppression method. Quantification limits were determined by linear regression of the obtained signal-to-noise ratios on product concentrations using a threshold of 6. **The obtained limits have been added to Table 2 in the revised manuscript, and the results have been further stressed on page 9, line 15**. In addition, **corresponding NMR spectra of the reference sample and regression results have been added as Supplementary Figs. 2 and 6, respectively**.

5. Can the authors please discuss the challenges and limitations of the technique for CO₂RR product analysis more? What can affect chemical shifts? What resolution (ppm) you have? What determines that? What are the experimental challenges and “details”? That is largely missing in the manuscript.

Chemical shifts of analytes are mainly affected by the pH and ionic strength of the sample. Neutral products like alcohols are less affected as DMSO (or DMSO₂) is added as an internal reference standard for chemical shift compensation. In the case of charged products, such as formate, chemical shifts are more strongly influenced by the pH or ionic strength. To illustrate, the observed resonances of acetone were 2.10 ppm for all investigated samples, while the

resonances of formate were in the range of 8.28-8.33 ppm, depending on the pH and concentration of the electrolyte. **We have added the observed and expected chemical shifts for the different products to Supplementary Table 1.**

The magnetic field strength of the spectrometer and the probe type generally determine the resolution in NMR spectroscopy. In our studies with a 500 MHz spectrometer equipped with a cryoprobe, we achieved digital resolutions of 0.001-0.005 ppm, depending on the signal of interest. Besides the main technical aspects of the spectrometer used, several other parameters affect the digital resolution of the final spectrum, e.g., magnetic field homogeneity (shimming), time-domain data points, acquisition time, or post-processing parameters. **A discussion on the digital resolution has been added on page 9, line 3. In addition, the corresponding technical details have now been included in the Methods section on page 24, lines 10.**

There are specific limitations of ^1H NMR spectroscopy as a quantification method in eCO_2RR . First of all, the product of interest must contain hydrogen atoms to be probed by electromagnetic radiation, which implies that possible products without hydrogen atoms cannot be detected and quantified, such as oxalate. Another limitation is the quantification of formaldehyde (without further treatment of the sample) with a chemical shift of 4.74 ppm (as methanediol in hydrated form), which overlaps with the water signal at 4.69 ppm (Hansen *et al. J. Phys. Chem. C* 126, 11026 (2022)).

Besides these limitations, several experimental challenges must be considered when applying ^1H NMR spectroscopy for eCO_2RR product analysis. One important consideration is the selection of a suitable internal standard for referencing and ensuring an accurate quantification. An internal standard of a known concentration should not interfere with analytes and provide a single resonance in the ^1H NMR spectrum. In this study, DMSO was chosen, but DMSO_2 also constitutes a possible internal standard. If no suitable compound is found, an external standard can be used as commonly employed in natural product analysis. Besides adding an internal standard to the sample, D_2O must be added for the field frequency lock, which is accompanied by a further dilution of the sample and, thus, increases detection and quantification limits. Furthermore, the suppression of the water signal is probably one of the most crucial experimental challenges, as only with an accurate suppression method the quantification of liquid products becomes viable. As described in the manuscript, the adapted WATERGATE suppression method in this study notably enhances signal-to-noise ratios allowing the quantification at lower concentration levels, as shown in Supplementary Fig. 1.

Therefore, **we have added a new subsection entitled “Experimental challenges and limitations of NMR spectroscopy as eCO_2RR product quantification method” to the Results and Discussion section, including a discussion on experimental challenges and limitations on pages 8-9.**

6. Please include a brief description of the error propagation calculation from Fig 1 also in the main text. Parts of the caption of Fig 1 can in fact move to the main text, to make it easier to understand when the authors refer to it.

Briefly, the errors were calculated based on a prospective quantification error of 5 ppm in the determination of product concentrations. *FEs* were then calculated assuming an Anderson-Schulz-Flory distribution with an α -value of 0.4 as observed for phosphate-derived Ni. Finally, the experimental error was propagated through the equation for calculating *FEs* (Supplementary Note 2) with a simplified Taylor series approximation resulting in electron-

number-dependent uncertainties. **Following the Reviewer's suggestion, we have included a description in the main text (page 3, line 22) and rephrased the caption of Fig. 1.**

7. Why do the authors choose to only correct for 80% of the solution resistance during the experiment? Why not 85%? 90%?

Preliminary tests were performed to identify the maximum compensation level without causing oscillations. Above 80%, strong current oscillations were observed hampering longer eCO₂RR experiments (>30 min), probably caused by a change in resistance over time and, ultimately, over-compensation. It is noteworthy that relatively large *iR* drops were present due to high current densities (up to 175 mA cm⁻²) and/or large solution resistances (up to 60 Ω). In addition, it is important to emphasize that the remaining uncompensated resistances were post-corrected to account for the entire *iR* drop. **We have now clarified in the Method (page 23, line 17) and Results and Discussion sections (page 17, line 9) that a real-time compensation of 80% was chosen to avoid current oscillations.**

8. The plural of pH is pH. Not pHs.

This has now been amended to pH values (page 19, line 5).

9. Please write down in the caption of Figures 5 and 6 what product is responsible for the other ~80-90% of faradaic efficiency (I assume H₂)?

The remaining Faradaic efficiencies is from H₂, as clarified in **Supplementary Table 2. This information is now included in the captions of Figs. 5 and 6 in the revised manuscript.**

10. In the conclusions the authors write "This work sets the practical basis for developing this new family of materials towards desired target products." However, the main product this catalyst make under CO₂ atmosphere is still hydrogen >90% for all conditions tested. This manuscript is nice as a "technical development" in the sense of the product detection methodology presented. However, as for the development of CO₂ electrocatalysts, I am missing a discussion as to how to stir the selectivity of these Ni-based catalysts, and how to improve the C-products faradaic efficiency. Without it, the statement in the conclusions is vague – what basis is being set for developing this new family of materials towards target products? As mentioned before: avoid overselling your work.

With the expanded range of products available from nickel-based electrocatalysts, it has become increasingly important to develop new quantification tools to enable adequate catalytic testing and facilitate experimental and theoretical efforts towards the development of catalyst design guidelines for this family of materials. In this regard, the analytical method presented in this manuscript is a seminal contribution towards this end.

We have illustrated the significance of our method by providing an initial analysis of performance trends and discussing their origin, facilitated by NMR analysis. To avoid any potential for misinterpretation of the main messages of the manuscript, we have revised the text to reinforce the technical aspects of the ¹H NMR method while clearly delimiting the early stage of understanding of these systems and the contribution of our analytical methods to future studies. **This has been highlighted in the 'Performance trends in carbon product formation on PD-Ni catalyst' section on page 19, line 17, and in the Conclusions section on page 20, line 24.**

Reviewer #2

Preikschas et al provide a carbon identification and quantification approach based on NMR Spectroscopy for analyzing the substrate mixture after CO₂ electroreduction over Ni catalysts as described in reference 5. The context of advanced analytics to this latter paper which has met much interest in the reception leave no doubt that this is a timely and relevant paper. I think, however, that the work would make for a much more compelling story that raises about the feel of additional analysis of previous work, if the following could be considered:

We acknowledge the valuable feedback provided by the Reviewer regarding the relevance of our contribution and appreciate their constructive criticism aimed at enhancing its impact. We have carefully acted upon all their suggestions.

1. The analytical part and the application part could be tied better together in my opinion, especially in the results section some transition from method to application would be desirable. In the application part, phrasing could be more accurate, for instance it cannot be any surprise that acetate is found in this reaction, albeit it has not been identified in reference 5.

This part culminates in speculation that is not discussed in depth, that bicarbonate and other buffer anions contribute beyond a pH effect, where bicarbonate serves as a hydrogen donor. Meaningful controls (if possible, at all) or computational work to corroborate the hypothesis would be desirable, as analogy to Au and Ag catalysts do not end the results section on a very rigorous note. A schematic figure of the suggested mechanistic effects including the newly identified chemical could be beneficial and improve impact. Overall and elsewhere, Figures are very nice.

I would prefer if the conclusion could end on a less generic note than "This work sets the practical basis for developing this new family of materials towards desired target products."- why not mention which specific questions have been addressed and which ones remain?

We appreciate the Reviewer for providing us with their valuable feedback, which is consistent with suggestions made by Reviewer #1. We acknowledge that, at this stage, the work's primary contribution is the development of a sensitive and versatile quantification method capable of assessing the performance of nickel-based electrocatalysts. Hence, we have revised our manuscript to focus on method development and refined our discussions on the observed performance trends and their origins. For instance, we have excluded the bicarbonate hypothesis and associated discussions, even though it has the potential to generate interest. However, conducting a deep treatment involving experimental and theoretical studies to substantiate the initial hypothesis is beyond the scope of this article. Nevertheless, we maintain that the observed performance trends are relevant to validate our quantification method's sensitivity. **We have highlighted these aspects in the 'Performance trends in carbon product formation on PD-Ni catalyst' section on page 19, line 17, and in the Conclusions section on page 20, line 24.**

2. This work is not least presented as an analytical toolbox, but many details are hard to judge, despite if Figure 3. The scripts are referred to as being available on github and the link should probably be repeated under (Supplementary Note 3). Does the toolbox provide a deconvolution and peak fitting in addition to integration and how are pH dependent shift changes addressed? Is the approach novel - similar approaches and toolboxes must exist in the metabolomics world? How is peak overlap addressed, for instance Figure 2 indicates that ethylene glycol will only yield overlapped signal?

The Automated Product Analysis Routine (APAR) will be constantly improved, and its code maintained. While the current version (v0.1.0) does not feature deconvolution and peak fitting, it is planned for the near future. A respective request has been pulled on the corresponding GitHub repository (<https://github.com/philpreikschas/apar>).

In the mentioned case of ethylene glycol, the singlet used for quantification at 3.54 ppm was well-separated from all peaks of ethanol's quartet centered at 3.53 ppm (not used for quantification), due to the relatively high resolution obtained on the 500 MHz spectrometer with a cryoprobe. However, we agree with the Reviewer that deconvolution and peak fitting functions are required to account for a prospective overlap of signals and to increase the quality of quantification results.

Changes in chemical shifts due to different pH values have been addressed *via* a user input of the expected pH value and the data table used for product identification containing product names, chemical shifts for different pH values, and coupling constants. **We have clarified the consideration of pH-dependent chemical shifts in Supplementary Note 3 and APAR's documentation on GitHub** (<https://github.com/philpreikschas/apar>).

Although a wide variety of scripts and toolboxes are available for handling NMR data and quantitative ^1H NMR analysis, none of these tools can be used for automated eCO₂RR product analysis. Most of the software is designed to handle one spectrum at a time (Pauli *et al.* *J. Nat. Prod.* 75, 834–851 (2012)) and, more importantly, are not able to identify eCO₂RR products as those do not include a chemical shift data table which is an integral component of APAR. Thus, we believe that APAR is the only available toolbox for a fully automated analysis of eCO₂RR products. **These considerations have been added to the 'Automated liquid product analysis routine' section on page 12, line 19.**

As the Reviewer commented that many details are hard to judge, **we have now created a detailed documentation of all parts and functions of APAR, which has been added to the respective GitHub repository** (<https://github.com/philpreikschas/apar>).

3. Experimental details about the NMR method remain a bit unclear to me:

A) Is the improved limit of detection the consequence of a bigger receiver gain owing to better water suppression? Is sensitivity really better than for GCMS? Could presaturation be optimized (e.g., on a standard, non-Cryoprobe instrument that may be accessible to most users) to allow for similar receiver gains as the WATERGATE experiment?

The improved detection limit indeed results from the full utilization of the dynamic receiver range due to better water suppression. We could not further optimize the water suppression by presaturation, as shown in Supplementary Fig. 1. Nevertheless, the adapted WATERGATE method does not require a spectrometer equipped with a cryoprobe and can also be used on standard instruments. We have successfully employed the adapted WATERGATE method on a 300 MHz spectrometer with a conventional BBFO probe. While the detection limits on a 300 MHz spectrometer are substantially higher than those on a 500 MHz spectrometer (as expected), the adapted WATERGATE method yielded sufficient results for quantifying main products with a concentration above 5-30 μM (SNRs > 6), which correlates to Faradaic efficiencies of approx. 1%. Therefore, **a comparison of spectra obtained on a 500 MHz (with cryoprobe) and a 300 MHz spectrometer has been added as Supplementary Fig. 5.** In addition, **quantification limits have been determined and added to Table 2 following the suggestion of Reviewer #1 (Comment 4).**

A general statement about whether the sensitivity of ^1H NMR spectroscopy with the presented method is better than that of a GC-MS system cannot be made. The sensitivity of a

GC system relies on several technical aspects, e.g., the selection of injector, columns, detector, carrier gas flows, and sample loop sizes, and can vastly differ depending on the product of interest. With the adapted WATERGATE method in our study, we identified four new products that could not be identified with a state-of-the-art GC instrument (probably due to its lower sensitivity). However, it should be noted that the sensitivity of ^1H NMR spectroscopy also depends on several parameters, e.g., magnetic field strength, probe type, magnet field homogeneity (shimming), or post-processing parameters. With the adapted WATERGATE method, we were able to quantify products in concentrations of 2-3 μM (SNRs > 20), which correspond to Faradaic efficiencies of 0.01% under typical reaction conditions (-1.2 V vs. RHE, 1 M KHCO_3). We believe that this level of sensitivity already matches the requirements of typical eCO_2RR investigations.

B) Acquisition times and recycle delays should be provided. Is it correctly understood that quantifications are achieved relative to internal DMSO by considering the six equivalent hydrogens from the internal standard? Can you show real spectra of the analyzed samples rather than solely the reference sample of SI1?

The acquisition time of all experiments was 3.3 s with recycle delays of 5 s. The Reviewer correctly understood that the quantification is based on the six equivalent hydrogen atoms of DMSO and the peak area of its single resonance at 2.60 ppm. **These essential details have been added to the Method section on page 24, line 10.** Furthermore, **representative spectra for all electrolytes used are now provided as Supplementary Fig. 18.**

C) How is the different relaxation behavior for different mixture components and the need to wait for $5 \cdot T_1$ accounted for in the quantification? How can you be sure that water suppression does not suppress signals near the water, for instance? Does the presence of paramagnetic species in samples need to be considered? And are samples neutralized before analysis to facilitate peak-ID?

We highly appreciate these precise questions helping profile our method. We agree that a relaxation delay of at least five times the longest T_1 is needed for accurate quantification. We chose 5 s based on reported literature (Kuhl *et al. Energy Environ. Sci.* 5, 7050 (2012), Adams *et al. Chem. Commun.* 49, 358 (2013), Bertheussen *et al. Catal. Today* 288, 54 (2017)). However, we have now considered more reports and realized that some of them report slower relaxations and, thereby, longer relaxation delays of up to 60 s (Chatterjee *et al., Dalton Trans.* 49, 4257 (2020), Hansen *et al. J. Phys. Chem. C* 126, 11026 (2022)). As a consequence of this discrepancy, we decided to perform additional ^1H NMR experiments to investigate the influence of different relaxation delays (1, 5, 30, 60 s), and we identified that a minimum of 60 s is required for full signal recovery. **These results on the signal recovery have been added as Supplementary Fig. 3.** Furthermore, **all samples have been re-measured with a relaxation delay of 60 s, and the respective Faradaic efficiencies have been updated in Figs. 5,6 and Supplementary Table 3.** For the highly volatile and less stable products, namely acetaldehyde, hydroxyacetone, acetone, *n*-propanol, and *i*-propanol, correction factors have been used to compensate for the relaxation effects. The correction factors of 1.00-1.18 have been obtained from the relaxation delay investigations at 5 and 60 s. Notably, the observed trends remained the same and were not affected by the partial signal recovery resembling a systematic error.

As the increased relaxation delays led to extended experiment times (75 min for 64 scans with 60 s), **we have investigated the applicability of an MRI contrast agent to lower the**

longitudinal relaxation times (T_1) and ultimately reduce experiment times. ProHance[®], a commercially-available Ga³⁺ contrast agent, has been chosen. However, the addition of 0.4 mM ProHance[®], as suggested in the literature (Hansen *et al. J. Phys. Chem. C* 126, 11026 (2022)), led to a serve increase in peak width from 0.98 to 1.45 Hz for the single resonance of formate at 8.33 ppm (Supplementary Fig. 7a) and other signals. Although this resolution might be adequate for quantifying products with well-separated signals, it is not applicable to complex product mixtures with overlapping ones as in this study. Therefore, a variation of ProHance[®]-concentration has been performed and its influence on the peak width investigated. It turn out that a concentration of 0.1 mM of ProHance[®] yields an almost unaltered peak width with 0.99 Hz (Supplementary Fig. 7b), sufficient to resolve overlapping peaks as exemplified for ethanol and *i*-propanol at 1.05-1.04 ppm (Supplementary Fig. 7c). Lastly, the signal recovery of different eCO₂RR has been investigated after the addition of 0.1 mM ProHance[®]. A recovery of 97% has been observed for all products with a 5 s relaxation delay, and full recovery has been achieved at 10 s (Supplementary Fig. 8). Therefore, the relatively long experiment times of 75 min (64 scans) could be substantially reduced by adding 0.1 mM ProHance[®] to 12 or 18 min, depending on the target degree of signal recovery. **These results have been added to the Results and Discussion section as a new subsection entitled “Lowering NMR experiment times with MRI contrast agents” on page 10. In addition, the corresponding data has been added as Supplementary Figs. 7 and 8.**

The adapted WATERGATE suppression method is advantageous when analyzing peaks close to the water signal and usually suppresses signals in a range of 4.6-4.8 ppm, as demonstrated by Adams and co-workers (*Chem. Commun.* 49, 358 (2013)). As indicated in Table 2, the signals considered for the quantification of liquid eCO₂RR products were well separated from the water signal without any resonances considered in the range of 3.99-8.33. Therefore, it can be excluded that the water suppression method influences the quantification results.

Paramagnetic species must be considered when working with Ni-based catalysts, and high-spin Ni²⁺ species might be present. However, post-reaction elemental analysis of the electrolytes *via* ICP-OES showed no indication of dissolved metal ions which could potentially influence the NMR analysis. Moreover, we have not experienced any contribution of paramagnetic species during NMR experimentations. **We have nonetheless added this relevant detail to the description of the method for future studies where nickel leaching into the electrolyte would be relevant in page 25, line 1.**

Lastly, samples have been measured as collected without any further treatment. The different chemical environments caused by the electrolytes used in our study lead to relatively small changes in chemical shifts, *e.g.*, 8.28-8.33 for formate. For this reason, a neutralization of the samples to facilitate peak identification was not required. As mentioned in the reply to Comment 2, the product identification of APAR is based on a data table including chemical shift data for different pH values.

*D) Is there any attempt to quantify formaldehyde or is the method agnostic to formaldehyde (see *e.g.*, ref 17)?*

Although the hydrated form of formaldehyde (methanediol) with a resonance at 4.74 ppm is close to the water signal (4.69 ppm), it should be detectable due to precise water suppression employed by the adapted WATERGATE method. Thus far, we have not observed any signal in the specific range during our experiments. **We have also performed control experiments by adding sodium bisulfite (NaHSO₃) to form a stable HCHO-bisulfite adduct with an**

altered chemical shift. These control experiments have not led to any signal indicating the presence of formaldehyde. **We have highlighted these aspects on page 8, line 16, and in the Method section on page 25, line 8.**

E) How is the presence of different forms accounted for, for instance acetaldehyde should be present in hydrate and aldehyde form, while only the hydrate form seems to be considered. Also, propionaldehyde likely may exist in a hydrate form, while here only the aldehyde form seems to be considered.

As indicated in Supplementary Table 1, we considered both forms (diol and aldehyde) in our quantification. However, we agree that we have made this point not sufficiently clear. Therefore, **we have now updated Table 2 and Supplementary Table 1 and indicated that both forms have been considered for quantification.**

F) Can you provide error estimates for your quantifications?

Repetition and preparation errors have been determined as standard deviations from three independent NMR measurements and two independent samples. The errors in the quantification of the most relevant products were below 2.2%. Furthermore, the repetitive analysis of the same sample was performed with a mean error of 1.5%, while the mean error in the determination of concentrations of two independently prepared samples was 0.9 %.

G) In Figure 2-why are there two NMR signals for methanol (I assume the OH would exchange quickly and only the methyl group is visible, also the shift seems not in accordance with hydroxy groups); probably "1H" Chemical shift should be named in the figure legend.

The chemical shift of methanol has now been amended. We apologize for this mistake, as methanol indeed shows only one signal due to the fast H/D exchange. In addition, **the title of the x-axis has been changed as requested.**

4. Some minor textual issues can be easily fixed: "till" throughout change to "until"; page 4 C₇H₁₄ should be C₇H₁₆.

The sentence containing the word "till" has been deleted during the revision. C₇H₁₄ has been corrected to C₇H₁₆ (page 4, line 21).

page 5 I am not sure if it is advisable to speculate on the analysis time of 20-60 min without the method, without any further substantiation.

The time estimate of 20-60 min has been deleted on page 5, line 10.

tempus: past tense throughout results section is encouraged.

The tense has been changed in two sentences on page 7, line 28 and page 8, line 4.

Ideally avoid using "This" without a noun; especially "This underlies fundamentally different reaction mechanisms for Cu and Ni in the activation of CO₂ and multicarbon product formation." becomes a suboptimal sentence, even if the meaning is understandable enough (This observation is consistent with previously suggested different reaction...)

The missing noun has been added to the referred sentence on page 18, line 22. No other dangling modifier has been found.

Ideally avoid starting sentences with "To illustrate," and "On the contrary" (By contrast)

Extra section for NMR data acquisition under methods is advised, including recycle delay, pH, experiment times, acquisition time of FID etc; maybe consider changing volumes from cm³ to mL.

We have added subsections on NMR data acquisition (page 24, line 10) and liquid product quantification (page 24, line 23) to the Methods section and moved the experimental details on NMR quantification from the catalyst evaluation subsection.

Table 1 could benefit from spaces between stoichiometric coefficients and chemicals/electrons
Spaces after coefficients have been added to all chemical equations in Table 1.

Reviewers' comments:

Reviewer #1 (Remarks to the Author):

I thank the authors for the modifications on the manuscript, however I still have one concern regarding the Faradaic Efficiencies (FE) reported.

The authors only provide the FE for carbon products in the main text (Fig. 5) and say that the remaining is H₂. However if I add up the FEs from Supplementary Tables 2 + 3 overall the FEs rarely reach 100%, mostly it is between 80-85% indicating an experimental issue with the product detection. The authors mention that for entry #1 the total FE = 71% "due to the low current density resulting in low product concentrations". However, also for higher currents the FEs do not reach 100%. If this is due to a limitation of the technique, OK, however, it has to be clearly disclosed and discussed in the main text by: (i) providing the total FEs in Fig. 5 next to the FE for carbon products, (ii) discussing why it does not reach 100% and what are the consequences of that.

You must understand that you cannot properly compare differences in FE for CO₂RR in different electrolytes when 20% of product detection are missing. For this "technical" paper this is OK, because you are not trying to have any deep mechanistic understanding of your selectivity, however if you want to apply this for systematic mechanistic studies, it is crucial to obtain 100±5% FEs.

I do not recommend this manuscript for publication unless this is addressed.

Another minor comment:

I strongly advise the authors to include "NMR" in their title.

Reviewer #2 (Remarks to the Author):

Preikschas et al have amended the manuscript thoroughly and in a largely satisfactory manner. Some additional input for consideration is the following:

1. Mostly, I fear that the novelty in terms of the NMR approach remains overstated. The method uses an NMR experiment described in reference 23 for analysis of electroreduction components and now also accounts for relaxation-derived errors as described in reference 16. Neither the use of reference compounds, solid water suppression schemes, perfect echo NMR sequences, relaxation agents etc are novel. I would encourage an additional thorough correction of sequences that could benefit from better use of references. For instance, page 7 largely describes findings that have been reported before (see e.g. <https://doi.org/10.1002/ange.201908006> and the reference 16 above; original watergate could be cited etc...)

2. The abstract indicates that the method is "-applicable to any catalyst-" where spaces around the hyphen would be recommended; more critically – is the statement true? Consider if homogeneous paramagnetic catalysts would obstruct the use of the ¹H NMR approach?

3. I concur that deconvolution is not a necessary part of the APAR approach, but it is actually state of the art and should be feasible using standard software such as Bruker Topspin and Mestrenova? I would suggest mentioning the term at least once in the manuscript.

4. Regarding the included data: The offset in SI Fig.4 is a bit unclear, 2 ppm offset for supposed formate is very big. Is the electrolyte correction described as good as can be? Page 4 in the SI refers to Fig S2a which does not exist. Is there a pH dependent offset on the lock substance, or why is the offset necessary? Does Mnova just calculate incorrectly, and is the use of such predictions really the best suited approach for unambiguous identifications?

Signals in Fig. S5 look rather unsymmetric, can that be amended by phasing/processing? It seems that the shim was rather poor, in fact in both instruments.

COMMSCHEM-23-0058A - Response to Reviewers

Comments in *blue* | Replies in black | Actions in **bold**.

Reference to figures, page and line numbers refer to the manuscript with highlighted changes.

Reviewer #1

I thank the authors for the modifications on the manuscript, however I still have one concern regarding the Faradaic Efficiencies (FE) reported.

The authors only provide the FE for carbon products in the main text (Fig. 5) and say that the remaining is H₂. However if I add up the FEs from Supplementary Tables 2 + 3 overall the FEs rarely reach 100%, mostly it is between 80-85% indicating an experimental issue with the product detection. The authors mention that for entry #1 the total FE = 71% "due to the low current density resulting in low product concentrations". However, also for higher currents the FEs do not reach 100%. If this is due to a limitation of the technique, OK, however, it has to be clearly disclosed and discussed in the main text by: (i) providing the total FEs in Fig. 5 next to the FE for carbon products, (ii) discussing why it does not reach 100% and what are the consequences of that.

You must understand that you cannot properly compare differences in FE for CO₂RR in different electrolytes when 20% of product detection are missing. For this "technical" paper this is OK, because you are not trying to have any deep mechanistic understanding of your selectivity, however if you want to apply this for systematic mechanistic studies, it is crucial to obtain 100±5% FEs.

I do not recommend this manuscript for publication unless this is addressed.

We thank the Reviewer for highlighting this aspect and placing their comment within the scope of this manuscript. With the target of this work being the development of a ¹H NMR protocol applicable to liquid product quantification, we acknowledge the importance of discussing the reasons behind non-closing charge balances in catalytic tests. Given that the developed NMR method can quantify products at concentrations equivalent to FEs of 0.01-0.08%, we can exclude that incomplete total FE is related to lack of sensitivity. Nonetheless, we fully understand the concern raised by the Reviewer and agree that the missing FEs and their origin need to be stressed within the manuscript and clearly separated from the capabilities of the NMR method. Therefore, **we have extensively discussed the reasons for the incomplete balance and its consequences directly next to the discussion of FEs based on the reasoning provided below (page 17, line 413) and highlighted it in the revised abstract.** In addition, **we have added the summed FEs to Fig. 5**, as requested by the Reviewer.

Closing the charge balance is a widespread challenge in the field of eCO₂RR. Aware of it, we applied state-of-the-art catalyst testing protocols (Larrazábal *et al. Acc. Mater. Res.* 2, 220 (2021), Diercks *et al. J. Electrochem. Soc.* 168, 064504 (2021), Ma *et al. Energy Environ. Sci.* 13, 977 (2020)). Importantly, we used the outlet flow from the electrochemical cell for quantification purposes to avoid overestimated FEs linked to the frequently used inlet flows (*e.g.*, Ji *et al. Nat. Catal.* 5, 251 (2022), Jin *et al. Nature* (2023) doi:10.1038/s41586-023-05918-8, Peng *et al. Nat. Commun.* 13, 1399 (2022)). While the total FEs in this study ranged from 71-95% based on the outlet flow, a recalculation with the inlet flow would yield FEs from 77 to 98%. FEs might thus not be directly comparable with literature due to different, non-standardized measurement procedures. Besides the limited comparability, we believe that the incomplete product balance is due to liquid product and (bi)carbonate crossover, dissolved

gaseous products and quantification of volatile compounds. These obstacles become even more pronounced for long-chain products as reflected in Figure 1 and briefly commented in the Introduction (page 3, line 46).

A bipolar membrane was used in this study, as they are claimed to inhibit crossover of products from the cathode to the anode, thereby preventing their oxidation and subsequent disruption of the carbon balance (Li *et al. Adv. Sustainable Syst.* 2, 1700187 (2018)). While the crossover could be substantially reduced compared to an anion exchange membrane, we still observed the crossing of formate and acetate, as pointed out in the Methods section (page 21, line 516). This finding is consistent with other studies (Ma *et al. Chem. Sci.* 11, 8854 (2020), Eriksson *et al. J. Electrochem. Soc.* 169, 034508 (2022)). Furthermore, it is important to acknowledge limitations in the quantification of gaseous and volatile products with GC systems currently employed in eCO₂RR research. Due to the large differences in product concentrations, a GC system with three or more detectors might be advised for enhanced sensitivity at lower currents. Moreover, dissolved gaseous products in the electrolyte and volatile oxygenates crossing the gas diffusion electrode cannot be assessed with current GC and experimental developments. Additional experimental features, such as gas-liquid separators and advanced GC systems capable of separating oxygenates and long-chain isomers, are needed for a complete carbon balance. It is worth noting that these functionalities have not yet been implemented in eCO₂RR experimental setups, and their detailed investigation falls beyond the scope of this manuscript.

Another minor comment:

I strongly advise the authors to include "NMR" in their title.

In this context, we fully agree with the Reviewer's suggestion to more clearly define the scope of the manuscript. Accordingly, **we have revised the title to 'NMR-based quantification of liquid products formed in CO₂ electroreduction on phosphate-derived nickel catalysts'**.

Reviewer #2

Preikschas et al have amended the manuscript thoroughly and in a largely satisfactory manner. Some additional input for consideration is the following:

We appreciate the Reviewer's positive appraisal of the revised manuscript and have thoroughly acted upon their additional suggestions.

1. Mostly, I fear that the novelty in terms of the NMR approach remains overstated. The method uses an NMR experiment described in reference 23 for analysis of electroreduction components and now also accounts for relaxation-derived errors as described in reference 16. Neither the use of reference compounds, solid water suppression schemes, perfect echo NMR sequences, relaxation agents etc are novel. I would encourage an additional thorough correction of sequences that could benefit from better use of references. For instance, page 7 largely describes findings that have been reported before (see e.g. <https://doi.org/10.1002/ange.201908006> and the reference 16 above; original watergate could be cited etc...)

We appreciate this comment, as we agree on the importance of clearly delimiting the novelty of our work. The main contribution of our study to NMR development is the integration and tailoring of certain NMR components from methodologies that have proven effective in other fields. It should be noted, however, that the mere combination of these elements does not yield a functional method. To illustrate this point, we have, for example, conducted an extended analysis of line broadening caused by relaxation agents, focusing on low concentrations of a wide range of products that exhibit overlapping signals. This analysis is crucial, as existing protocols are impractical for complex product mixtures, as observed in the case of phosphate-derived Ni catalysts. The overall outcome of this design process is the development of an NMR protocol that achieves quantification limits approximately one order of magnitude lower than those previously reported, without introducing additional complexity. **In response to the Reviewer's suggestion, we have carefully revised the language in the manuscript to ensure that any claims of novelty are appropriately qualified (see page 5, line 82, page 6, line 111, page 7, line 151, page 7, line 153, and page 10, line 468).**

In accordance, we have ensured that appropriate referencing is provided to the original sources. Specifically, **we have included references to the original work on WATERGATE** (Ref. 25; Sklenar *et al. J. Magn. Reson. A* 102, 241 (1993)) as recommended, wherever the WATERGATE method with perfect echo sequences utilized in this study is cited. Furthermore, to contextualize our results on relaxation errors and agents, **we have expanded the discussion incorporating previously reported studies, including the suggested references (page 7, line 143, page 10, line 227, and page 10, line 229).**

2. The abstract indicates that the method is “-applicable to any catalyst-” where spaces around the hyphen would be recommended; more critically – is the statement true? Consider if homogeneous paramagnetic catalysts would obstruct the use of the 1sH NMR approach?

We acknowledge that without further clarification, it may not be evident that this claim pertains to any liquid product distribution obtained from known supported electrocatalysts. **To address this concern, we have replaced the phrase ‘any catalyst’ with ‘any supported catalyst’.**

3. I concur that deconvolution is not a necessary part of the APAR approach, but it is actually state of the art and should be feasible using standard software such as Bruker Topspin and Mestrenova? I would suggest mentioning the term at least once in the manuscript.

As remarked by the Reviewer, deconvolution and peak fitting are not necessary processing steps for product quantification. However, we agree that these are features enhancing the capabilities of APAR that were in fact planned for later versions. In attention to the Reviewer's comment, we have developed peak fitting and deconvolution functions based on the Imfit Python library, which features non-linear least-squares minimization and curve fitting (Newville *et al.* Zenodo 10.5281/zenodo.7887568 (2023)). **We have now added peak fitting and convolution to APAR, revised APAR's description on in Supplementary Note 3, and updated Fig. 3 accordingly.**

4. Regarding the included data: The offset in SI Fig.4 is a bit unclear, 2 ppm offset for supposed formate is very big. Is the electrolyte correction described as good as can be?

To clarify the quality of prediction for the alleged case of formate, we have created the following Figure R1 illustrating the offset between experimental and predicted values.

Figure R1 | Chemical shift predictions after electrolyte compensation.

As reported in Table 1, the experimentally obtained ^1H chemical shift of formate is 8.33 ppm. The prediction made using the Mnova software suite in D_2O yielded a value of 9.41 ppm, corresponding to an offset of 1.08 ppm. After correction based on linear regression, as described in the Supplementary Discussion, the chemical shift of formate was recalculated as 8.86 ppm, leading to a 0.53 ppm deviation. Overall, this approach reduced the relative mean squared error between observed and calculated values, which fell from 10.6 to 6.67%. Please see below for the discussion on electrolyte correction.

Page 4 in the SI refers to Fig S2a which does not exist.

The reference to Fig. S4a has now been corrected on page 4 in the Supplementary Information. We apologize for this oversight.

Is there a pH dependent offset on the lock substance, or why is the offset necessary? Does Mnova just calculate incorrectly, and is the use of such predictions really the best suited approach for unambiguous identifications?

Regarding the need for this correction, it is noteworthy that the Mnova software suite is limited to five different solvents. Consequently, the prediction was conducted in D₂O (Supplementary Discussion). Given the systematic offset observed in the corresponding parity plot (Fig. S4a), we deduced that this offset is likely attributed to the pH and ionic strength of the electrolyte rather than a pH-induced shift of the lock substance (D₂O). The Mnova software suite employs a so-called ensemble predictor (Wang *et al. Renewable Sustainable Energy Rev.* 75, 796 (2017)), which combines different prediction and machine learning algorithms trained on extensive datasets (Cobas. *Magn. Reson. Chem.* 58, 512 (2020)). While the predictive power of this approach has not been validated in a general sense, we believe such validation to be beyond the scope of this study. It is crucial to clarify that these predictions were not used in isolation for product identification. Instead, they aided in the selection of appropriate reference samples, which formed the basis for the experimental validation of all identified products. **In the Supplementary Discussion, we have explicitly stated that NMR predictions were not relied upon as the sole means of product identification.**

Signals in Fig. S5 look rather unsymmetric, can that be amended by phasing/processing? It seems that the shim was rather poor, in fact in both instruments.

In Fig. S5, it is observed that some signals exhibit a slight tailing; however, it is important to note that this tailing does not have any practical impact on the integration and quantification results. As correctly pointed out by the Reviewer, the tailing could be attributed to a slight inhomogeneity in the magnetic field (shimming, Z⁴). Shimming is a procedure that should be performed by the individual user prior to conducting an NMR experiment. Therefore, the quality of the shimming process is completely independent of and unrelated to the methodology presented in this manuscript.

One aspect contributing to this issue in our case is the utilization of the automated shimming algorithm available in the Bruker TopSpin software, which was employed on the 300 MHz spectrometer equipped with an autosampler for routine measurements. Additionally, semi-automatic shimming was thoroughly performed on the 500 MHz spectrometer, incorporating a maximum shim function order of 8 (CryoProbe). Further optimization of the shimming quality could, however, not be achieved, which we attribute to the relatively high salt concentrations present in the samples, as reference samples in pure D₂O or other deuterated solvents did not behave similarly. **Following the Reviewer's suggestion, we have conducted additional data processing (manual phasing, apodization) to maximize the quality of the spectra within these limitations, and accordingly updated Fig. S5.** Despite these adjustments, the integration results remained nearly unchanged, with only a small deviation of 0.5%.

REVIEWERS' COMMENTS:

Reviewer #2 (Remarks to the Author):

I believe that the manuscript is sufficiently improved by now with a more balanced phrasing. The only thing that I would be hesitant about remains the very absolute statement that the method is "applicable to any supported catalyst" as leaching supported catalysts for instance would not be great and as I do not see why the statement is necessary at all. I think this is a minor semantic matter that can be agreed upon on author/editor level. Congratulations to the nice manuscript.

COMMSCHEM-23-0058B - Response to Reviewers

Comments in *blue* | Replies in black | Actions in **bold**.

Reference to figures, page and line numbers refer to the manuscript with highlighted changes.

Reviewer #2

I believe that the manuscript is sufficiently improved by now with a more balanced phrasing. The only thing that I would be hesitant about remains the very absolute statement that the method is "applicable to any supported catalyst" as leaching supported catalysts for instance would not be great and as I do not see why the statement is necessary at all. I think this is a minor semantic matter that can be agreed upon on author/editor level. Congratulations to the nice manuscript.

We appreciate the Reviewer's positive feedback on the revised manuscript. In response to their suggestion, **the statement 'applicable to any supported catalyst' has been removed from the abstract.**